# MITIGATING STRATEGY-SELECTION BIAS IN REASONING FOR MORE EFFECTIVE TEST-TIME SCALING

## ABSTRACT

Test-time scaling (TTS) has been shown to improve the performance of large language models (LLMs) by sampling and aggregating diverse reasoning paths. However, existing research has overlooked a critical issue: *selection bias of reasoning strategies during scaling*. Specifically, when generating reasoning processes, LLMs tend to follow certain strategies (*e.g.,* algebraic solutions for math problems) while neglecting other valid alternatives (*e.g.,* geometric solutions), resulting in insufficient exploration of the solution space. To further understand the impact of this bias, we present a theoretical analysis that reveals when it undermines the effectiveness of test-time scaling. Motivated by this theoretical insight, we introduce TTS-Uniform, a framework designed to mitigate the selection bias of reasoning strategies. It (i) identifies potential strategies, (ii) uniformly allocates the sampling budget across them, and (iii) filters out unstable strategies prior to aggregation. Experimental results show that TTS-Uniform significantly enhances scaling effectiveness across multiple mainstream LLMs and benchmark datasets.

## 1 INTRODUCTION

Chain-of-thought (CoT) (Wei et al., 2022; Kojima et al., 2022) enhances the reasoning capabilities of large language models (LLMs) by explicitly unfolding intermediate steps (*i.e.,* reasoning paths) before arriving at the final answer. Building on CoT, test-time scaling (TTS) (Zhang et al., 2025; Ji et al., 2025) further improves performance by sampling and aggregating diverse paths. However, existing TTS research (Wang et al., 2022; Snell et al., 2024) overlooks a critical limitation of CoT, which in turn constrains the effectiveness of scaling. Specifically, when tackling a problem, CoT reasoning tends to follow certain strategies while neglecting other valid alternatives. For example, a mathematical problem may be solvable through either an algebraic or a geometric method, yet LLMs often exhibit a strong preference for the algebraic approach. We refer to this phenomenon as **strategy-selection bias**. Such bias leads scaling to disproportionately concentrate its sampling on a narrow subset of preferred strategies, leaving large portions of the solution space unexplored and thus reducing the overall diversity of reasoning paths, as illustrated in Figures 1 and 2.

To further examine the impact of strategy-selection bias on TTS, we conduct a theoretical analysis. We partition the available strategies for a given problem into two categories: low-complexity and high-complexity, according to the minimum number of tokens required for an LLM to solve the problem. The analysis reveals that when the skewed dominant strategies fall into the low-complexity category, such bias can unexpectedly improve performance by concentrating the sampling budget on simpler reasoning paths. In contrast, when high-complexity strategies dominate, their longer and less reliable paths diminish the performance gains that would otherwise be achieved through scaling.

To mitigate the scaling instability induced by strategy-selection bias, we propose a novel TTS framework, termed **TTS-Uniform**. This framework begins by extracting potential solution strategies for a given problem, either through coarse-grained (conceptual-level) or fine-grained (step-level) equivalence. Each extracted strategy is then appended to the original problem text and fed into the LLM, thereby guiding the model to reason along the specified strategy. On this basis, TTS-Uniform uniformly allocates the sampling budget across all strategies to ensure adequate exploration of the solution space. It then estimates the uncertainty of each strategy by calculating the entropy of the answers generated from their respective sampled reasoning paths. Strategies with high uncertainty

are discarded, as they are likely to belong to high-complexity ones. Finally, the answers from the remaining strategies are aggregated via majority voting to produce the final prediction.

The proposed TTS-Uniform maintains the benefits of scaling when simple reasoning strategies dominate, while dynamically reallocating the budget toward simpler reasoning paths when complex ones are prevalent, thereby effectively enhancing overall TTS performance. Building on this foundation, the main contributions of this paper can be summarized as follows:

- We identify and formalize strategy-selection bias in CoT reasoning, showing that LLMs favor a narrow set of strategies while neglecting other valid alternatives.
- We partition reasoning strategies by their minimum token requirement and theoretically analyze how bias toward different categories affects test-time scaling.
- We propose a novel TTS framework that extracts potential solution strategies, uniformly allocates the budget, and discards unstable ones to counteract strategy-selection bias.

## 2 STRATEGY-SELECTION BIAS IN LLM REASONING

To rigorously quantify the phenomenon of strategy-selection bias, we first formalize the concept of distinct reasoning strategies by defining an equivalence relation over correct reasoning paths, thereby grouping semantically equivalent approaches into discrete strategy classes.

**Definition 2.1** (Reasoning Strategy Set). *Let $\mathcal{Q}$ be a problem, and let $\mathcal{R}$ denote the set of all correct reasoning paths that solve $\mathcal{Q}$. Define an equivalence relation $\sim$ on $\mathcal{R}$. For $r, r' \in \mathcal{R}$, $r \sim r'$ if and only if $r$ and $r'$ have equivalent semantics. Assume the quotient set $\mathcal{R}/\sim$ is finite and write its $m$ equivalence classes as: $\mathcal{R}_1, \ldots, \mathcal{R}_m, [m] := \{1, \ldots, m\}$. We refer to each $\mathcal{R}_i$ as a reasoning strategy for $\mathcal{Q}$, and define the reasoning strategy set as:*

$$\mathcal{S} := \{\mathcal{R}_1, \ldots, \mathcal{R}_m\}.$$

*If a reasoning path $r$ belongs to the $i$-th class, we write $r \in \mathcal{R}_i$.*

Given Definition 2.1, once the potential reasoning strategies for a problem are identified, a natural question arises: *How does an LLM distribute its sampling probability across these reasoning strategies?* In the absence of inherent preference, an ideal solver would sample approximately uniformly from $\mathcal{S}$, thereby ensuring full coverage of the solution space.

To investigate the above question, we conducted an experiment on mathematical problems that admit multiple valid solutions. Specifically, we randomly sampled from the AIME dataset and used GPT-4o-mini to generate 500 independent reasoning paths for each. These paths were then analyzed using two complementary approaches: (i) encoding each path with a sentence encoder (MiniLM (Wang et al., 2020)) to obtain embeddings and constructing the corresponding t-SNE visualization, and (ii) clustering the paths using another more powerful reasoning model (OpenAI-o3). Partial results are shown in Figures 1 and 2, with additional results provided in Appendix.

We observed a consistent pattern across all samples: whether clustering was performed directly on the embeddings or through a more powerful reasoning model, the proportion of reasoning paths following the dominant strategy was consistently much larger than the combined proportion of all other strategies, often exceeding twice their total. This indicates that the distribution of reasoning strategies generated by GPT-4o-mini deviates significantly from a uniform distribution. We formally refer to this phenomenon as strategy-selection bias in LLM reasoning, defined as follows.

**Definition 2.2** (Strategy-Selection Bias). *Let $\mathcal{Q}$ be a problem whose reasoning strategy set is $\mathcal{S} := \{\mathcal{R}_1, \ldots, \mathcal{R}_m\}$, as defined in Definition 2.1. Input $\mathcal{Q}$ into an LLM yields a distribution over $\mathcal{R}_i$. Consequently, there's also an underlying distribution over $\mathcal{S}$ denoted as $p$:*

$$p(\mathcal{R}_i) := \Pr_{r \sim \text{LLM}(Q)}\big[r \in \mathcal{R}_i\big], \qquad i \in [m],$$

*where $p$ can be estimated through repeated querying of an LLM with $\mathcal{Q}$. The pair $(\mathcal{S}, p)$ is said to exhibit strategy-selection bias in LLM reasoning if:*

$$D(p \parallel U) > \delta,$$

*where $U(\mathcal{R}_i) = \frac{1}{m}$ is the uniform reference distribution, $D(\cdot \parallel \cdot)$ is a statistical divergence measure (e.g., KL divergence), and $\delta > 0$ is a predefined significance threshold.*

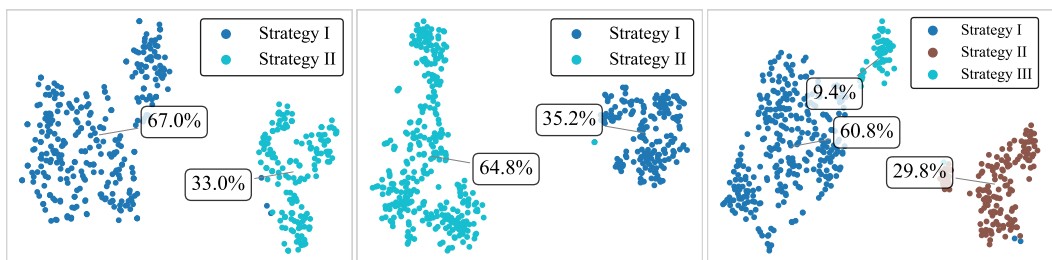

Figure 1: t-SNE visualization of embeddings (MiniLM encoder) for the 500 independent reasoning paths generated by GPT-4o-mini on three randomly selected AIME samples.

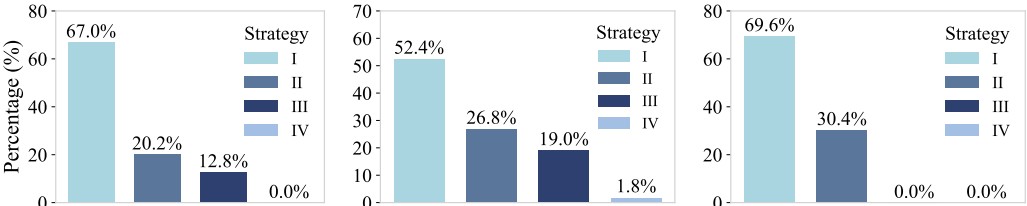

Figure 2: Percentage distribution of reasoning strategy clusters identified by OpenAI-o3 for the 500 independent reasoning paths generated by GPT-4o-mini on three randomly selected AIME samples.

## 3 EFFECTS OF STRATEGY-SELECTION BIAS ON TEST-TIME SCALING

As discussed in Section 2, existing LLMs exhibit a pronounced skew toward a small subset of dominant reasoning strategies rather than sampling uniformly from the full strategy set $\mathcal{S}$. Such skewed distributions can undermine the effectiveness of the test-time scaling paradigm (Zhang et al., 2025), which aims to maximize predictive performance under a fixed computational budget (*e.g.,* a limited number of sampled reasoning paths) during inference. To investigate this impact, we first introduce several foundational definitions and an assumption (Section 3.1), and then build on this foundation to theoretically analyze how bias influences test-time scaling (Section 3.2).

### 3.1 COMPLEXITY AND ERROR CORRELATION IN REASONING STRATEGIES

We begin by introducing the minimum token requirement of a strategy as a measure of its complexity, which we then use to characterize the likelihood of errors when an LLM employs that strategy.

**Definition 3.1** (Minimum Token Requirement of the Strategy). *For a given problem $\mathcal{Q}$ and its associated reasoning strategy $\mathcal{R}_i$, the minimum token requirement, denoted $\ell_i$, is defined as the smallest number of tokens necessary to execute all essential reasoning steps required to derive the correct solution to $\mathcal{Q}$, without introducing any redundant information.*

**Assumption 3.2** (Complexity-Error Correlation). *We assume that the minimum token requirement $\ell_i$ characterizes the complexity of executing strategy $\mathcal{R}_i$ for problem $\mathcal{Q}$. Higher complexity (i.e., a larger $\ell_i$) entails a higher probability of erroneous output when an LLM employs $\mathcal{R}_i$.*

This assumption follows from the fact that reasoning in LLMs relies on the step-by-step generation of tokens, each carrying a non-zero error probability. As the required reasoning sequence lengthens ($\ell_i$ increases), these errors can amplify and accumulate, significantly increasing the likelihood of an incorrect final outcome. This phenomenon is consistent with findings reported in prior studies (Zeng et al., 2025) and is further corroborated by our empirical results in Appendix A.5.

Leveraging Definition 3.1 and Assumption 3.2, we partition the reasoning strategy set $\mathcal{S}$ into two disjoint subsets according to reasoning complexity, as defined below.

**Definition 3.3** (Strategies of Low and High Complexity). *Given a problem $\mathcal{Q}$, an associated set of strategies $\mathcal{S} := \{\mathcal{R}_1, \ldots, \mathcal{R}_m\}$, and their corresponding complexities $\mathcal{L} = \{\ell_1, \ldots, \ell_m\}$, let $\tau$ be a predefined complexity threshold. A reasoning strategy $\mathcal{R}_i \in \mathcal{S}$ is classified as a* low-complexity *strategy for $\mathcal{Q}$ if $\ell_i < \tau$; otherwise ($\ell_i \geq \tau$), it is classified as a* high-complexity *strategy.*

## 3.2 THEORETICAL ANALYSIS

After partitioning reasoning strategies into low- and high-complexity groups (Definition 3.3), we investigate how the effectiveness of test-time scaling is influenced when high-frequency (dominant) strategies are biased toward one of these complexity categories. To formalize this, we define the complexity skew of a strategy distribution in terms of how its probability mass is concentrated over strategies of varying complexity. In particular, we introduce a partial ordering over distributions to capture the extent to which a distribution favors simpler versus more complex strategies.

Let $X$ and $Y$ be random variables representing reasoning strategies, taking values in the reasoning strategy set $\mathcal{S}$. Their respective distributions are denoted $X \sim p_X$ and $Y \sim p_Y$, where $p_X(\mathcal{R}_i) = \Pr[X = \mathcal{R}_i]$ and $p_Y(\mathcal{R}_i) = \Pr[Y = \mathcal{R}_i]$ for $i \in [m]$. Since each strategy $\mathcal{R}_i$ is uniquely associated with its index $i$, we represent strategies by their indices, so that $X$ and $Y$ take values in $[m]$, with $\Pr[X = i] = p_X(i)$ and $\Pr[Y = i] = p_Y(i)$.

Under Assumption 3.2, higher-complexity strategies are associated with higher error rates. Let $\epsilon_i := \Pr(\text{LLM errs using } \mathcal{R}_i) \in [0, 1]$ denote the error rate of strategy $\mathcal{R}_i$, and relabel the strategies such that $\epsilon_1 \le \epsilon_2 \le \cdots \le \epsilon_m$. Hence, smaller indices correspond to strategies with lower complexity.

To compare the degree of skew between two distributions, we introduce the following ordering. Formally, we say that $X$ exhibits a stronger preference for low-complexity strategies than $Y$ if, for all $x \in \mathbb{R}$, it holds that $\Pr[X > x] \le \Pr[Y > x]$, denotes as $X \preceq Y$. Intuitively, under this ordering, the smaller random variable concentrates more probability on low-complexity strategies. Since $X$ and $Y$ are discrete random variables taking values in $[m]$, the condition $\forall x, \Pr[X > x] \le \Pr[Y > x]$ is equivalent to $\forall i \in [m-1], \sum_{j>i} p_X(j) \le \sum_{j>i} p_Y(j)$.

Next, we examine the expected error rate under each distribution. Define the error rate function $\text{err} : [m] \to [0, 1]$, where $\text{err}(i) = \epsilon_i$. Then, the expected error rates under distributions $p_X$ and $p_Y$ are $\mathbb{E}_{X \sim p_X}[\text{err}(X)]$ and $\mathbb{E}_{Y \sim p_Y}[\text{err}(Y)]$, respectively. The central question is whether the ordering $X \preceq Y$ guarantees a lower expected error. This is established by the following result.

**Lemma 3.4.** $X \preceq Y$ *if and only if for any non-decreasing function* $u : \mathbb{R} \to \mathbb{R}$ *such that* $\mathbb{E}[|u(X)|] < \infty$ *and* $\mathbb{E}[|u(Y)|] < \infty$, *we have* $\mathbb{E}[u(X)] \le \mathbb{E}[u(Y)]$.

Since the domain of the function $\text{err}(\cdot)$ can be extended to $\mathbb{R}$ while preserving its non-decreasing property without altering the expected error, Lemma 3.4 immediately yields the desired conclusion:

**Theorem 3.5.** *If* $X \preceq Y$, *then* $\mathbb{E}_{X \sim p_X}[\text{err}(X)] \le \mathbb{E}_{Y \sim p_Y}[\text{err}(Y)]$.

Based on Theorem 3.5[1], we derive the following key conclusions about the impact of strategy-selection bias on test-time scaling: (i) when the skewed dominant strategies belong to the low-complexity group, scaling is generally more effective compared to uniform strategy sampling; (ii) conversely, when these strategies are of high complexity, the effectiveness of scaling is reduced. These conclusions are further corroborated by empirical results presented in Appendix A.4.

These findings motivate the development of a new framework specifically designed to enhance the effectiveness and reliability of test-time scaling, as introduced in the next section.

## 4 METHODOLOGY

We propose a novel approach, called TTS-Uniform (see Figure 3), to mitigate the instability of test-time scaling induced by strategy-selection bias. Specifically, we first extract potential solution strategies from the given problem to construct a reasoning strategy set (see Section 4.1). Each strategy is then sequentially appended to the problem text and individually fed into the LLM, aiming to steer its reasoning along the specified strategy. To ensure balanced sampling, we allocate an equal number of samples to each strategy by dividing the total sampling budget by the number of strategies, thereby inducing a uniform sampling distribution over the strategy set. Finally, we identify and filter out reasoning paths that likely originate from high-complexity strategies, and aggregate the remaining paths to produce the final prediction (see Section 4.2).

---

[1]The proofs of Lemma 3.4 and Theorem 3.5 are provided in Appendix A.2.

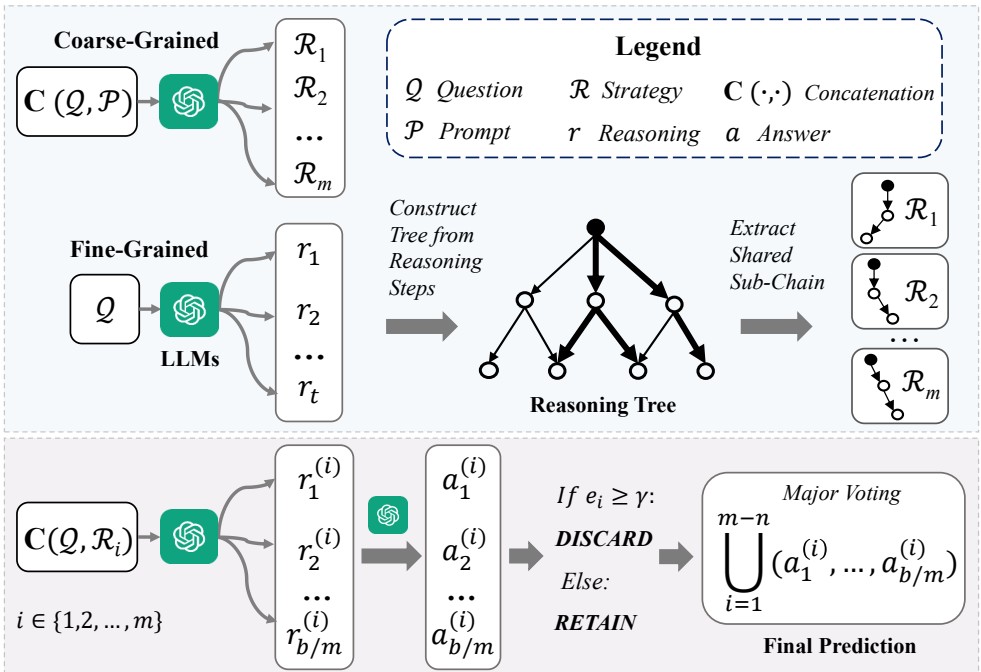

Figure 3: The flowchart of the proposed TTS-Uniform consists of two main parts, *i.e.,* **Strategy Extraction** (light blue block) and **Strategy Aggregation** (light purple block). Strategy extraction is performed via coarse-grained equivalence (prompting the LLM to list high-level conceptual strategies) or fine-grained equivalence (constructing a reasoning tree from sampled steps and extracting shared sub-chains). Strategy aggregation allocates a uniform sampling budget across strategies, discards high-entropy (complex) ones, and applies majority voting to obtain the final prediction.

## 4.1 STRATEGY EXTRACTION

Given a problem $\mathcal{Q}$ and an LLM employed to solve it, our objective is to leverage the model to extract all possible solution strategies. The primary challenge lies in establishing clear distinctions between strategies. In other words, it concerns how to characterize the semantic equivalence (see Definition 2.1) between different reasoning paths within the same strategy.

To address this challenge, we propose two categories of equivalence relations to accommodate different distinction criteria, *i.e.,* coarse-grained equivalence and fine-grained equivalence. Specifically, *coarse-grained equivalence* categorizes strategies based on abstract, high-level perspectives. Under this equivalence class, reasoning paths of the same strategy may exhibit local variations in specific procedural aspects while maintaining a consistent underlying reasoning logic. Conversely, *fine-grained equivalence* differentiates strategies through discrepancies in concrete reasoning steps. Within this class, distinct paths of the same strategy demonstrate high similarity at the step-execution level, with potential identity within acceptable tolerance thresholds.

### 4.1.1 COARSE-GRAINED EQUIVALENCE

To derive the reasoning strategy set based on coarse-grained equivalence, we design a prompt $\mathcal{P}$ that guides the LLM away from directly solving problem $\mathcal{Q}$. Instead, the prompt instructs the LLM to identify and summarize high-level reasoning strategies potentially applicable to $\mathcal{Q}$. The prompt is constructed as follows: **#** $\mathcal{P}$*: What are the abstract solution strategies for the above problem? Please identify all potential approaches at a conceptual level without providing computational details.*

The problem $\mathcal{Q}$ is concatenated with $\mathcal{P}$ and fed into the LLM to generate the strategy set:

$$\{\mathcal{R}_1, \ldots, \mathcal{R}_m\} = \textbf{Extract}(\textbf{LLM}(\textbf{Concat}(\mathcal{Q}, \mathcal{P}))), \tag{1}$$

where the **Concat**$(\cdot)$ denotes the sequential concatenation of the specified text segments. The **Extract**$(\cdot)$ defines a rule for segmenting text from responses based on specified criteria.

In addition, this method can be naturally extended to a few-shot setting. Specifically, the prompt $\mathcal{P}$ in Eq. (1) can be augmented with a set of carefully selected high-quality exemplars that demonstrate how to abstract solution strategies for similar problems. This enables the LLM to perform effective in-context learning and potentially yield more accurate and diverse strategy sets.

### 4.1.2 FINE-GRAINED EQUIVALENCE

Although reasoning strategies derived from coarse-grained equivalence are suitable for most datasets in practice, they become less effective when high-level conceptual categories are difficult to define or ambiguous. In such challenging cases, we resort to fine-grained equivalence, which operates at the level of individual reasoning steps, enabling strategy differentiation based on concrete procedural variations rather than abstract conceptual distinctions.

To achieve this, we first allocate a small portion of the budget $t$ (where $t < b$) from the total computational budget $b$. We then draw $t$ independent samples from the LLM for the given problem $\mathcal{Q}$, thereby obtaining a set of exemplar reasoning paths:

$$\{r_1, \ldots, r_t\} = \{\textbf{LLM}(\mathcal{Q})_j \mid j = 1, 2, \ldots, t\}, \tag{2}$$

where $\textbf{LLM}(\cdot)_j$ denotes the $j$-th sampling result produced by the LLM.

Subsequently, we utilize the set $\{r_1, \ldots, r_t\}$ to construct a reasoning tree, which characterizes the diverse reasoning strategies that may be employed to answer the problem $\mathcal{Q}$. Specifically, we treat $\mathcal{Q}$ as the root node and decompose each reasoning path $r_k = (s_1^{(k)}, s_2^{(k)}, \ldots, s_{n_k}^{(k)})$, where $s_i^{(k)}$ denotes the $i$-th reasoning step in the $k$-th path, into a sequence of fine-grained logical steps. Each step is represented as a node, and the nodes are sequentially connected to form the path $\mathcal{Q} \to s_1^{(k)} \to s_2^{(k)} \to \cdots \to s_{n_k}^{(k)}$. When identical reasoning steps appear in different paths, they are merged into a single node in order to explicitly capture the structural commonalities across paths. Collectively, all reasoning paths form a multi-branch reasoning tree $\mathcal{T}_\mathcal{Q}$, which is formally defined as:

$$\mathcal{T}_\mathcal{Q} = \langle \mathcal{V}, \mathcal{E} \rangle, \tag{3}$$

where the vertex set is defined as the set of all unique nodes in the reasoning tree:

$$\mathcal{V} = \{\mathcal{Q}\} \cup \bigcup_{k=1}^{t} \left\{ s_i^{(k)} \mid i = 1, \ldots, n_k \right\}, \tag{4}$$

and the edge set is given by the set of directed links between consecutive nodes:

$$\mathcal{E} = \bigcup_{k=1}^{t} \left\{ (\mathcal{Q}, s_1^{(k)}), (s_1^{(k)}, s_2^{(k)}), \ldots, (s_{n_k-1}^{(k)}, s_{n_k}^{(k)}) \right\}. \tag{5}$$

Next, we further extract potential reasoning strategies $\{\mathcal{R}_1, \ldots, \mathcal{R}_m\}$ from the constructed reasoning tree $\mathcal{T}_\mathcal{Q}$ in an automated manner. Specifically, this is achieved by identifying ***shared sub-chains*** across different reasoning paths, enabling the aggregation of recurring logical patterns. Each shared sub-chain can be regarded as a potential reasoning strategy, representing recurring logical structures or patterns of thought that appear across multiple reasoning paths.

### 4.2 STRATEGY AGGREGATION

After obtaining the reasoning strategy set described on Section 4.1, each strategy is appended to the original problem and subsequently fed into the LLM. This process is designed to encourage the LLM to adhere to the specified strategy during its reasoning. For each strategy, we sample $b/m$ reasoning paths (or $(b-t)/m$ paths in the case of coarse-grained strategies) as follows:

$$\{r_1^{(i)}, \ldots, r_{b/m}^{(i)}\} = \{\textbf{LLM}(\textbf{Concat}(\mathcal{Q}, \mathcal{R}_i))_j \mid j = 1, 2, \ldots, b/m\}, \tag{6}$$

where $r_k^{(i)}$ denotes the $k$-th reasoning path under the $i$-th strategy.

To enable more effective aggregation of the reasoning paths obtained from Eq. (6), we prune those that are likely guided by high-complexity strategies, as such paths are more prone to errors.

To this end, we first estimate the complexity of each strategy. A straightforward approach is to use the average token length of the generated reasoning paths under each strategy as a proxy (see Definition 3.1 and Assumption 3.2). However, empirical observations (see Appendix A.7) reveal that this average often deviates significantly from the true minimum token requirement (*i.e.,* the intrinsic complexity) of the strategy. This discrepancy primarily arises because LLMs tend to introduce substantial repetition or redundancy in intermediate steps when generating reasoning paths, resulting in actual token consumption that far exceeds the minimal amount required to complete the reasoning. Consequently, estimates based on average length are generally unreliable.

To address the above issues, we propose a more realistic complexity estimation method. Intuitively, complex strategies are more likely to destabilize the reasoning process due to error accumulation and amplification, which results in higher uncertainty in the final predictions. Based on this insight, we introduce the information entropy of final predictions across sampled reasoning paths as a surrogate measure of strategy complexity[2]. Specifically, for each strategy $\mathcal{R}_i$, we collect the final predictions generated from its sampled reasoning paths:

$$\{a_1^{(i)}, \ldots, a_{b/m}^{(i)}\} = \{\textbf{LLM}(\textbf{Concat}(\mathcal{Q}, \mathcal{R}_i, r_j^{(i)})) \mid j = 1, 2, \ldots, b/m\}. \tag{7}$$

We then compute the entropy of the answer distribution for strategy $\mathcal{R}_i$:

$$e_i = \mathcal{H}(\mathcal{R}_i) = -\sum_{a \in \mathcal{A}_i} P_i(a) \log P_i(a), \tag{8}$$

where $\mathcal{A}_i$ denotes the set of all final answers produced under $\mathcal{R}_i$, and $P_i(a)$ represents the empirical frequency of answer $a$. A higher entropy value indicates greater uncertainty in the predictions, implying that the strategy is less stable and potentially more complex. Therefore, we discard the top $n$ strategies with the highest entropy values, yielding a filtered strategy set $\{\mathcal{R}_1, \ldots, \mathcal{R}_{m-n}\}$.

Finally, the final prediction $\hat{a}$ for question $\mathcal{Q}$ is obtained by aggregating the predictions from all reasoning paths associated with the remaining strategies:

$$\hat{a} = \text{MajorVote}\left(\bigcup_{i=1}^{m-n} a_1^{(i)}, \ldots, a_{b/m}^{(i)}\right), \tag{9}$$

where $\text{MajorVote}(\cdot)$ denotes the majority voting function used to aggregate the answers.

Finally, TTS-Uniform mitigates test-time scaling instability via two mechanisms: (i) when simple strategies dominate, entropy filtering removes inadvertently introduced high-complexity paths, preserving scaling benefits; (ii) when complex strategies dominate, uniform sampling and entropy filtering jointly enhance the proportion of low-complexity paths, thereby improving performance.

## 5 EXPERIMENTS

Our proposed Uniform-TTS framework includes three variants: (i) Uniform-C (Z), a coarse-grained zero-shot prompting approach; (ii) Uniform-C (F), which adds few-shot exemplars to coarse-grained prompting; and (iii) Uniform-F, a fine-grained method that constructs a reasoning tree and extracts shared sub-chains to form strategies. We evaluate them on three reasoning benchmarks: AQuA (Ling et al., 2017), AIME 2024 (MAA, 2024), and AIME 2025.

We adopt GPT-4o-mini (weaker) and GPT-4.1-mini (stronger) as base models, and assess performance with Pass@$k$ (Chen et al., 2021) and Acc@$k$. Pass@$k$ checks if at least one correct answer appears among $k$ samples, while Acc@$k$ measures majority-vote accuracy. We set $k = \{32, 64\}$ for GPT-4o-mini and $k = \{16, 32\}$ for GPT-4.1-mini. For comparison, we include two parallel test-time scaling methods, Self-Consistency (Wang et al., 2022) and PromptC-SC (Wu et al., 2025a), in which computations are performed independently, and one sequential method, ARI (Wu et al., 2025b), where subsequent computations build on earlier results.

---

[2]Appendix A.5 presents the relationship among strategy complexity (*i.e.,* minimum token requirement), entropy of answer distributions from paths guided by the strategy, and the proportion of correct answers.

| Dataset | Method | GPT-4o-mini | | | | GPT-4.1-mini | | | |
|---|---|---|---|---|---|---|---|---|---|
| | | **P@32** | **P@64** | **A@32** | **A@64** | **P@16** | **P@32** | **A@16** | **A@32** |
| **AQuA** | Self-Consistency | 95.3 | 96.1 | 89.0 | 89.0 | 94.9 | 96.5 | 88.6 | 88.6 |
| | PromptC-SC | 96.9 | 97.6 | 89.0 | 89.3 | 96.9 | 97.6 | 89.8 | 88.6 |
| | ARI | 94.0 | 94.0 | 89.3 | 89.3 | 91.7 | 91.7 | 90.6 | 90.6 |
| | Uniform-C (Z) | $96.1_{0.8}$ | $97.6_{1.5}$ | $89.0_{0.0}$ | $89.0_{0.0}$ | $96.5_{1.6}$ | $\mathbf{98.4}_{1.9}$ | $90.6_{2.0}$ | $90.6_{2.0}$ |
| | Uniform-C (F) | $\mathbf{98.0}_{2.7}$ | $\mathbf{98.4}_{2.3}$ | $89.8_{0.8}$ | $89.8_{0.8}$ | $\mathbf{96.9}_{2.0}$ | $97.6_{1.1}$ | $89.8_{1.2}$ | $90.2_{1.6}$ |
| | Uniform-F | $94.5_{0.8}$ | $97.2_{1.1}$ | $\mathbf{89.8}_{0.8}$ | $\mathbf{89.8}_{0.8}$ | $96.8_{1.9}$ | $97.2_{0.7}$ | $\mathbf{91.3}_{2.7}$ | $\mathbf{91.3}_{2.7}$ |
| **AIME (2024)** | Self-Consistency | 33.3 | 43.3 | 10.0 | 10.0 | 83.3 | 86.6 | 60.0 | 60.0 |
| | PromptC-SC | 46.6 | 56.6 | 13.3 | 13.3 | 83.3 | 86.6 | 60.0 | 63.3 |
| | ARI | 26.6 | 30.0 | 13.3 | 20.0 | 70.0 | 70.0 | 63.3 | 63.3 |
| | Uniform-C (Z) | $40.0_{6.7}$ | $\mathbf{60.0}_{16.7}$ | $16.6_{6.6}$ | $20.0_{10.0}$ | $83.3_{0.0}$ | $83.3_{3.3}$ | $66.6_{6.6}$ | $66.6_{6.6}$ |
| | Uniform-C (F) | $40.0_{6.7}$ | $53.3_{10.0}$ | $\mathbf{20.0}_{10.0}$ | $20.0_{10.0}$ | $76.6_{6.7}$ | $\mathbf{86.6}_{0.0}$ | $70.0_{10.0}$ | $70.0_{10.0}$ |
| | Uniform-F | $\mathbf{50.0}_{16.7}$ | $50.0_{6.7}$ | $16.6_{6.6}$ | $16.6_{6.6}$ | $\mathbf{83.3}_{0.0}$ | $83.3_{3.3}$ | $\mathbf{73.3}_{13.3}$ | $\mathbf{73.3}_{13.3}$ |
| **AIME (2025)** | Self-Consistency | 23.3 | 23.3 | 3.3 | 3.3 | 73.3 | 73.3 | 53.3 | 56.6 |
| | PromptC-SC | 33.3 | 33.3 | 13.3 | 13.3 | 73.3 | 76.6 | 60.0 | 60.0 |
| | ARI | 23.3 | 23.3 | 13.3 | 13.3 | 73.3 | 76.6 | 66.6 | 66.6 |
| | Uniform-C (Z) | $33.3_{10.0}$ | $40.0_{16.7}$ | $16.6_{13.3}$ | $20.0_{16.7}$ | $80.0_{6.7}$ | $\mathbf{83.3}_{10.0}$ | $63.3_{10.0}$ | $63.3_{6.7}$ |
| | Uniform-C (F) | $36.6_{13.3}$ | $\mathbf{43.3}_{20.0}$ | $\mathbf{23.3}_{20.0}$ | $20.0_{16.7}$ | $\mathbf{80.0}_{6.7}$ | $80.0_{6.7}$ | $\mathbf{70.0}_{16.7}$ | $70.0_{13.4}$ |
| | Uniform-F | $\mathbf{36.6}_{13.3}$ | $40.0_{16.7}$ | $13.3_{10.0}$ | $20.0_{16.7}$ | $73.3_{0.0}$ | $80.0_{6.7}$ | $66.6_{13.3}$ | $66.6_{10.0}$ |

Table 1: Performance of methods on three datasets using GPT-4o-mini and GPT-4.1-mini. P@ and A@ denote Pass@ and Acc@, respectively. Bold indicates the best result per dataset and model. Subscripts indicate improvements over Self-Consistency (red = improvement, green = decrease).

| Model (Dataset) | Self-Consistency | Uniform-C (Z) | Uniform-C (F) | Uniform-F | Avg. | Model Avg. |
|---|---|---|---|---|---|---|
| **GPT-4o-mini (AIME 2024)** | 0.138 | 0.040 | 0.020 | 0.030 | 0.057 | 0.053 |
| **GPT-4o-mini (AIME 2025)** | 0.112 | 0.020 | 0.012 | 0.053 | 0.049 | |
| **GPT-4.1-mini (AIME 2024)** | 0.089 | 0.012 | 0.010 | 0.009 | 0.030 | 0.031 |
| **GPT-4.1-mini (AIME 2025)** | 0.074 | 0.030 | 0.012 | 0.013 | 0.032 | |

Table 2: Divergence between the sampled strategy distribution and the uniform distribution.

## 5.1 MAIN RESULTS

From Table 1, we observe that all three variants of our proposed Uniform-TTS framework achieve stable and significant improvements over baseline methods across datasets and model settings. For example, on the AQuA dataset, Uniform-F attains the highest accuracies with GPT-4o-mini and GPT-4.1-mini (*i.e.,* Acc@64=89.8% and Acc@32=91.3%), substantially surpassing Self-Consistency. This demonstrates that, rather than repeatedly relying on a single dominant reasoning path, balanced strategy sampling can effectively enhance both accuracy and stability.

On the most challenging AIME datasets, the advantages of Uniform-TTS become even more pronounced. For instance, on AIME 2024 with GPT-4o-mini, our three variants outperform the three comparison methods by an average of 11.1% in Pass@64 and 4.5% in Acc@64. This indicates that more complex problems often admit a wider variety of solution strategies, and Uniform-TTS is able to exploit this diversity to achieve larger performance gains. Moreover, Uniform-TTS provides strong benefits for weaker reasoning LLMs. For example, on AIME 2025 with GPT-4o-mini, Uniform-C (F) improves Acc@32 from 3.3% (Self-Consistency) to 23.3%, representing a 20.0% absolute increase. For the stronger GPT-4.1-mini, the improvements are smaller in magnitude but remain consistent. For instance, Uniform-C (Z) increases Acc@32 from 56.6% to 63.3%. We will therefore further analyze this phenomenon in the next section.

## 5.2 BIAS COMPARISON ACROSS MODELS WITH DIFFERENT REASONING ABILITIES

To further investigate the phenomenon observed in Section 5.1, namely that weaker reasoning models benefit more from the our method than stronger ones, we conducted a detailed analysis of their bias levels. Specifically, for GPT-4o-mini and GPT-4.1-mini, we generated 100 independent reasoning paths per question across two datasets and evaluated four methods in total, including one baseline

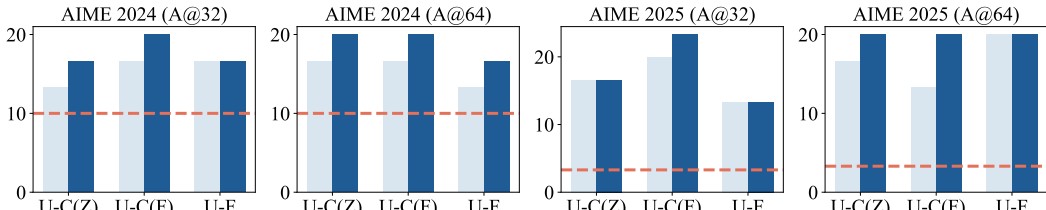

Figure 4: Ablation study of TTS-Uniform on AIME 2024 and AIME 2025 using GPT-4o-mini. The red dashed line denotes Self-Consistency obtained by removing all modules. The light blue bars correspond to strategy extraction with uniform sampling only (w/o entropy filtering), while the dark blue bars correspond to the full model with all three modules, including entropy-based filtering.

method (Self-Consistency) and three variants of our proposed TTS-Uniform. We then employed a more powerful reasoning model (OpenAI-o3) to cluster these reasoning paths and computed the frequency of each cluster. Based on these results, we calculated the divergence between the strategy sampling distribution and the uniform distribution, as defined in Definition 2.2, thereby quantifying the degree of bias. The experimental results are presented in Table 2.

The findings reveal that, on average across the two datasets, the divergence produced by Self-Consistency on GPT-4o-mini is 53.4% higher than that on GPT-4.1-mini. This indicates that weaker reasoning models exhibit substantially stronger bias compared with their more capable counterparts. This phenomenon can be attributed to the fact that weaker models are more susceptible to the inherent bias in pre-training data, and the maximum likelihood training objective further reinforces such bias, causing them to replicate high-frequency solution patterns and rely heavily on dominant strategies. In contrast, stronger models, equipped with greater reasoning capacity and broader world knowledge, can leverage longer reasoning chains to escape these high-frequency paradigms and explore alternative valid strategies, thereby mitigating strategy-selection bias.

## 5.3 ABLATION STUDY

To evaluate the contribution of each component in the proposed TTS-Uniform framework, we conduct ablation experiments on the AIME 2024 and AIME 2025 datasets using GPT-4o-mini. The framework consists of three key modules: strategy extraction, uniform sampling, and entropy-based filtering. We compare three configurations: (i) removing all modules, which degenerates into the self-consistency baseline Wang et al. (2022); (ii) applying strategy extraction and uniform sampling but discarding entropy-based filtering; and (iii) keeping all modules intact.

From Figure 4, we observe that the self-consistency mehthod consistently yields the lowest performance across all settings, confirming that naively aggregating multiple reasoning paths without considering strategy diversity brings limited benefit. Incorporating strategy extraction with uniform sampling (case (ii)) yields notable improvements, highlighting the importance of allocating the sampling budget evenly across diverse strategies. Finally, adding entropy-based filtering (case (iii)) delivers further gains, particularly on AIME 2025 (*e.g.,* A@32 increases from 20.0% to 23.3% under Uniform-C (F)), demonstrating that discarding unstable or high-entropy strategies enhances robustness. Moreover, Uniform-C (F), which integrates all three modules, consistently achieves the strongest results, outperforming both the baseline and partial variants.

## 6 CONCLUSION

This paper highlights the often overlooked issue of strategy-selection bias in chain-of-thought reasoning during test-time scaling (TTS). Our theoretical and empirical findings suggest that the effectiveness of scaling critically depends on the relationship between bias and the complexity of reasoning paths. Beyond proposing TTS-Uniform as a practical remedy, these insights offer broader implications for future research. In particular, our findings suggest the urgent need for designing training and alignment procedures that cultivate balanced exploration of diverse reasoning strategies, rather than relying solely on test-time interventions. More generally, addressing strategy-selection bias may further open new directions for improving performance in other reasoning paradigms, such as multi-step planning, program synthesis, and multimodal inference.

## REPRODUCIBILITY STATEMENT

The experimental settings are described at the beginning of Section 5. To further support reproducibility, we provide the complete code implementation in **Supplementary Materials**. A README file is included, which contains step-by-step instructions for quickly running the code and reproducing the reported experiments.

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

## A APPENDIX

### A.1 STATEMENT ON LLM ASSISTANCE

Large language models (LLMs) were used solely for grammar checking and language polishing. They were not involved in the conception of the research problem, the design of methodology, the analysis of results, or the generation of any research content.

### A.2 OMITTED PROOFS AND SUPPLEMENTARY DEFINITIONS

#### A.2.1 PROOF OF LEMMA 3.4 AND THEOREM 3.5

We introduce the notion of an *upper set* to provide an alternative necessary and sufficient condition for $X \preceq Y$.

**Definition A.1** (Upper Set). *A set $U \subseteq \mathbb{R}$ is called an **upper set** if, for any $x \in U$, every $y > x$ also belongs to $U$.*

In fact, every upper set must take the form of either $(a, +\infty)$ or $[a, +\infty)$ for some $a \in \mathbb{R}$.

**Lemma A.2.** *$X \preceq Y$ if and only if $\mathbb{E}[\mathbb{1}_U(X)] \leq \mathbb{E}[\mathbb{1}_U(Y)]$ for all upper sets $U$, where $\mathbb{1}_U(\cdot)$ is the indicator function of $U$, i.e.,*

$$\mathbb{1}_U(x) = \begin{cases} 1, & x \in U, \\ 0, & otherwise. \end{cases}$$

*Proof.* Let $F_X(z) = \Pr[X \leq z]$ and $F_Y(z) = \Pr[Y \leq z]$ be the cumulative distribution functions (CDFs) of $X$ and $Y$, respectively. Since any upper set $U$ must be of the form $(z, +\infty)$ or $[z, +\infty)$ for some $z \in \mathbb{R}$, the condition of the lemma is equivalent to the following two inequalities holding for all $z \in \mathbb{R}$:

1. $\mathbb{E}[\mathbb{1}_{(z,+\infty)}(X)] \leq \mathbb{E}[\mathbb{1}_{(z,+\infty)}(Y)]$

2. $\mathbb{E}[\mathbb{1}_{[z,+\infty)}(X)] \leq \mathbb{E}[\mathbb{1}_{[z,+\infty)}(Y)]$

By definition, $X \preceq Y$ is equivalent to $\Pr[X > z] \leq \Pr[Y > z]$ for all $z \in \mathbb{R}$. For any interval $U \subseteq \mathbb{R}$, we have

$$\mathbb{E}[\mathbb{1}_U(X)] = \int_{-\infty}^{+\infty} \mathbb{1}_U(x) \mathrm{d}F_X(x)$$
$$= \int_U \mathrm{d}F_X(x)$$
$$= \Pr[X \in U]$$

Thus, we have $\mathbb{E}[\mathbb{1}_{(z,+\infty)}(X)] = \Pr[X > z]$ and $\mathbb{E}[\mathbb{1}_{[z,+\infty)}(X)] = \Pr[X \geq z]$. Therefore, the lemma is equivalent to proving that the statement "$\forall z \in \mathbb{R}, \Pr[X > z] \leq \Pr[Y > z]$" is equivalent to the following two conditions holding simultaneously:

(i) $\forall z \in \mathbb{R}, \Pr[X > z] \leq \Pr[Y > z]$

(ii) $\forall z \in \mathbb{R}, \Pr[X \geq z] \leq \Pr[Y \geq z]$

First, we note that the proof for the $\Leftarrow$ direction is trivial, as (i) is part of the condition. For the $\Rightarrow$ direction, we need to show that $\forall z \in \mathbb{R}, \Pr[X > z] \leq \Pr[Y > z]$ implies $\forall z \in \mathbb{R}, \Pr[X \geq z] \leq \Pr[Y \geq z]$. To prove this, we utilize the following property of probability measures.

**Lemma A.3** (Continuity of Probability, Theorem 1.1.1 in Durrett (2019)). *Let $\{A_n\}_{n=1}^{\infty}$ be a sequence of events such that $A_1 \supseteq A_2 \supseteq \cdots$. If $A = \bigcap_{n=1}^{\infty} A_n$, then $\lim_{n \to \infty} \Pr[A_n] = \Pr[A]$.*

Since the event $\{X \geq z\}$ can be expressed as the intersection of a decreasing sequence of events, i.e., $\{X \geq z\} = \bigcap_{n=1}^{\infty} \{X > z - \frac{1}{n}\}$ (this holds because $\bigcap_{n=1}^{\infty} \left(z - \frac{1}{n}, +\infty\right) = [z, +\infty)$), it follows from Lemma A.3 that

$$\Pr[X \geq z] = \lim_{n \to \infty} \Pr\left[X > z - \frac{1}{n}\right].$$

By assumption, for any positive integer $n$, we have

$$\Pr\left[X > z - \frac{1}{n}\right] \leq \Pr\left[Y > z - \frac{1}{n}\right].$$

Taking the limit as $n \to \infty$ on both sides, we obtain

$$\Pr[X \geq z] \leq \Pr[Y \geq z].$$

This completes the proof. $\qquad\square$

Now we are ready to prove lemma 3.4. We restate lemma 3.4 here.

**Lemma A.4** (Lemma 3.4, restated). *$X \preceq Y$ if and only if, for any non-decreasing function $u : \mathbb{R} \to \mathbb{R}$ with $\mathbb{E}[|u(X)|] < \infty$ and $\mathbb{E}[|u(Y)|] < \infty$, it holds that $\mathbb{E}[u(X)] \leq \mathbb{E}[u(Y)]$.*

*Proof.* The direction $\Leftarrow$ is trivial, since for any upper set $U$, the function $\mathbb{1}_U(x)$ is non-decreasing. One only needs to take $u(x) = \mathbb{1}_U(x)$, and the desired conclusion follows immediately from the lemma A.2.

We now prove the direction $\Rightarrow$. We will need the layer cake representation.

**Lemma A.5** (Layer Cake Representation, Lieb & Loss (2001)). *Let $f$ be a non-negative measurable real-valued function defined on a measure space $(\Omega, \mathcal{A}, \mu)$. Then*

$$f(x) = \int_0^{+\infty} \mathbb{1}_{\{y \in \Omega : f(y) \geq t\}}(x)\mathrm{d}t.$$

We also require the following fact.

**Fact A.6.** *Every monotone function is measurable.*

Fact A.6 holds because any monotone function has at most countably many points of discontinuity. In what follows, we will assume all functions under discussion are measurable and omit this qualifier.

We first consider the case where $u(x)$ is bounded below. Without loss of generality, we may assume $u(x) \geq 0$, since if $u(x)$ is bounded below, adding a sufficiently large constant (which does not affect the expectation) makes it non-negative. When the measure space is a probability space $(\mathbb{R}, \mathcal{A}, \Pr)$, for a random varible $X$ with $\mathbb{E}[|X|] < \infty$, Lemma A.5 and the Fubini–Tonelli theorem imply:

$$\mathbb{E}[u(X)] = \int_{-\infty}^{+\infty} u\mathrm{d}\Pr$$

$$= \int_0^{+\infty} \Pr[\{y \in \mathbb{R} : u(y) \geq t\}]\mathrm{d}t$$

$$= \int_0^{+\infty} \mathbb{E}[\mathbb{1}_{U_t}(X)]\mathrm{d}t,$$

where it is easy to verify that $U_t = \{y \in \mathbb{R} : u(y) \geq t\}$ is an upper set. By Lemma A.2, we have:

$$X \preceq Y \Rightarrow \forall U, \ \mathbb{E}[\mathbb{1}_U(X)] \leq \mathbb{E}[\mathbb{1}_U(Y)]$$

$$\Rightarrow \mathbb{E}[u(X)] = \int_0^{+\infty} \mathbb{E}[\mathbb{1}_{U_t}(X)]\mathrm{d}t \leq \int_0^{+\infty} \mathbb{E}[\mathbb{1}_{U_t}(Y)]\mathrm{d}t = \mathbb{E}[u(Y)].$$

This completes the proof for the case where $u(x)$ is non-negative.

Next, we consider the case where $u(x)$ is not bounded below. Define $u_n(x) = \max\{u(x), -n\}$. Then for any fixed positive integer $n$, $u_n(x) \geq -n$, which is lower-bounded. By the previous result, we have:

$$X \preceq Y \Rightarrow \mathbb{E}[u_n(X) + n] \leq \mathbb{E}[u_n(Y) + n] \Rightarrow \mathbb{E}[u_n(X)] \leq \mathbb{E}[u_n(Y)].$$

We now apply the monotone convergence theorem.

**Lemma A.7** (Monotone Convergence Theorem, Theorem 1.6.6 in Durrett (2019)). *Let $\{X_n\}_{n=1}^{\infty}$ be a sequence of non-negative random variables converging almost surely to $X$, i.e., $X_n \xrightarrow{\text{a.s.}} X$. Then $\lim_{n \to \infty} \mathbb{E}[X_n] = \mathbb{E}[X]$.*

Since $u(x)$ is non-decreasing, the sequence $\{u_n(x)\}$ has a uniform lower bound (independent of $n$), and there exists $x'$ such that for all $x > x'$, $u_n(x) = u(x)$. Thus, for the random variable $X$, $u_n(X) \xrightarrow{\text{a.s.}} u(X)$. By Lemma A.7, for sufficiently large $M$, taking limits on both sides of the inequality $\mathbb{E}[u_n(X) + M] \leq \mathbb{E}[u_n(Y) + M]$ and then subtracting $M$, we obtain:

$$\mathbb{E}[u(X)] \leq \mathbb{E}[u(Y)],$$

which completes the proof. $\qquad\square$

Now we are ready to prove theorem 3.5.

**Theorem A.8** (Theorem 3.5, restated). *If $X \preceq Y$, then $\mathbb{E}[\mathrm{err}(X)] \leq \mathbb{E}[\mathrm{err}(Y)]$.*

*Proof.* To apply lemma 3.4, we need to modify the $\mathrm{err}$ function slightly. Define $\widetilde{\mathrm{err}} : \mathbb{R} \to [0, 1]$ such that

$$\widetilde{\mathrm{err}}(x) = \begin{cases} e_1, & x \leq 1 \\ e_i, & x \in (i, i+1], \ i = 1, 2, \cdots, m-2 \\ e_m, & x > m-1 \end{cases}$$

It is easy to verify that $\widetilde{\mathrm{err}}(i) = \mathrm{err}(i)$, so we can directly compare $\mathbb{E}[\widetilde{\mathrm{err}}(X)]$ and $\mathbb{E}[\widetilde{\mathrm{err}}(Y)]$. $\widetilde{\mathrm{err}}(x)$ is a step function on $\mathbb{R}$, which is clearly non-decreasing. Thus, applying lemma 3.4 under the condition $X \preceq Y$, we have $\mathbb{E}[\widetilde{\mathrm{err}}(X)] \leq \mathbb{E}[\widetilde{\mathrm{err}}(Y)]$, and therefore $\mathbb{E}[\mathrm{err}(X)] \leq \mathbb{E}[\mathrm{err}(Y)]$. $\qquad\square$

### A.2.2 First-Order Stochastic Dominance

The order relation ($X \preceq Y$) used in the main text to measure distributional preference is known in economics and probability theory as *First-Order Stochastic Dominance (FOSD)*. This criterion provides a precise mathematical definition for the notion that "one random variable yields better outcomes than another in a probabilistic sense." According to the standard definition, $Y$ is said to first-order stochastically dominate $X$ if the cumulative distribution function $F_Y$ is everywhere no greater than $F_X$, i.e., $F_Y(z) \leq F_X(z)$ for all $z \in \mathbb{R}$. Thus, the notation $X \preceq Y$ in this paper is equivalent to $Y$ exhibiting first-order stochastic dominance over $X$.

**Definition A.9** (First-Order Stochastic Dominance). *For random variables $X, Y$ on $\mathbb{R}$, we say $Y$ has FOSD over $X$ if $\forall z \in \mathbb{R}$, $\Pr[X > z] \leq \Pr[Y > z]$, denoted $X \leq_{\mathrm{FOSD}} Y$.*

### A.2.3 Proof of Theorem 3.5 in the Discrete Case

We may note that Lemma 3.4 is both elegant and concise, applicable to both continuous and discrete random variables. However, its proof can appear somewhat intricate. In fact, the central result of this paper, Theorem 3.5, is established under the assumption of discrete random variables, namely the condition $\forall i \in [m-1], \sum_{j>i} p_X(j) \leq \sum_{j>i} p_Y(j)$. Under this assumption, the proof becomes considerably simpler. We provide the argument here. To streamline notation, let $X, Y$ be random variables representing inference strategies, taking values only in $[m]$, with probabilities denoted $x_i = p_X(i)$ and $y_i = p_Y(i)$ for $i \in [m]$. The corresponding expected error rates of strategies $X$ and $Y$ are given by $\mathbb{E}[\mathrm{err}(X)] = \sum_{i \in [m]} \epsilon_i x_i$ and $\mathbb{E}[\mathrm{err}(Y)] = \sum_{i \in [m]} \epsilon_i y_i$.

**Theorem A.10.** *Suppose $0 \leq \epsilon_1 \leq \cdots \leq \epsilon_m \leq 1$, if for every $1 \leq i \leq m-1$, it holds that $\sum_{i < j \leq m} x_j \leq \sum_{i < j \leq m} y_j$, then $\sum_{i \in [m]} \epsilon_i x_i \leq \sum_{i \in [m]} \epsilon_i y_i$.*

*Proof.* Since $\sum_{i \in [m]} x_i = \sum_{i \in [m]} y_i = 1$, the condition $\sum_{i < j \leq m} x_j \leq \sum_{i < j \leq m} y_j$ for all $1 \leq i \leq m-1$ is equivalent to $\sum_{j \leq i} (x_j - y_j) \geq 0$ for all $1 \leq i \leq m$. To prove $\sum_{i \in [m]} \epsilon_i x_i \leq \sum_{i \in [m]} \epsilon_i y_i$,

it suffices to show that $\sum_{i \in [m]} \epsilon_i(x_i - y_i) \le 0$. Define $w_i = x_i - y_i$ and set $\sum_{j < 1} w_j = 0$, then

$$
\begin{aligned}
\sum_{i=1}^{m} \epsilon_i(x_i - y_i) &= \sum_{i=1}^{m} \left( \sum_{j \le i} w_j - \sum_{j < i} w_j \right) \\
&= \sum_{i=1}^{m} \left( \sum_{j \le i} w_j \right) \epsilon_i - \sum_{i=1}^{m-1} \left( \sum_{j \le i} w_j \right) \epsilon_{i+1} \\
&= \epsilon_m \underbrace{\sum_{i=1}^{m} w_i}_{=0} - \sum_{i=1}^{m-1} \left( \sum_{j \le i} w_j \right) (\epsilon_{i+1} - \epsilon_i) \quad \text{(Abelian transformation)} \\
&= \sum_{i=1}^{m-1} \underbrace{\sum_{j \le i} w_j}_{\ge 0, \, \forall i \in [m]} \underbrace{(\epsilon_i - \epsilon_{i+1})}_{\le 0} \\
&\le 0.
\end{aligned}
$$

This completes the proof. $\qquad\square$

### A.3 RELATED WORK

#### A.3.1 CHAIN-OF-THOUGHT REASONING

Chain-of-thought (CoT) reasoning (Wei et al., 2022) guides large language models (LLMs) to explicitly generate intermediate steps, thereby improving interpretability and reasoning quality. CoT has achieved notable success in diverse domains, including mathematical problem solving (Sprague et al., 2024) and multi-modal inference (Chen et al., 2024; Mitra et al., 2024; Wang et al., 2025b), where its transparent reasoning paths also enable collaborative and verifiable inference.

Research on CoT reasoning has primarily centered on prompt construction (Shao et al., 2023). Manual prompting methods such as CCoT (Chia et al., 2023) achieve strong performance but require substantial human effort and suffer from poor generalizability. Automated methods like Auto-CoT (Zhang et al., 2022) reduce manual costs yet often introduce instability. To balance performance and usability, semi-automatic prompting approaches such as AutoMate CoT (Shum et al., 2023) have been proposed. Despite these advances, existing CoT methods typically encourage models to favor certain reasoning trajectories, which can inadvertently restrict the diversity of solution strategies. This limitation becomes particularly critical in the context of scaling.

#### A.3.2 TEST-TIME SCALING

Test-time scaling (TTS) extends CoT reasoning by allocating additional inference-time computation to sample and aggregate diverse reasoning paths. Recent surveys provide systematic taxonomies of TTS methods, covering what, how, where, and how well to scale (Zhang et al., 2025), and interpret its role as a transition from fast System-1 style intuition to deliberate System-2 style reasoning (Ji et al., 2025). Others highlight its significance for cognition engineering, positioning TTS as a key driver of deeper reasoning in LLMs (Xia et al., 2025; Chen et al., 2025).

Methodological innovations focus on enhancing efficiency and robustness. For instance, MetaScale adaptively selects cognitive strategies through evolving meta-thoughts (Liu et al., 2025), while s1-32B demonstrates that scaling effects can be achieved with minimal training data and a simple budget forcing mechanism (Muennighoff et al., 2025). At the same time, limitations of naive scaling have been exposed. (Zeng et al., 2025) observe that longer chains of thought may reduce accuracy because of excessive self-revisions, and (Wang et al., 2025a) establish theoretical upper bounds showing diminishing returns once the compute budget exceeds a certain threshold.

Although these works improve the efficiency and stability of TTS, they do not explicitly consider strategy-selection bias, where LLMs concentrate on preferred solution patterns while ignoring al-

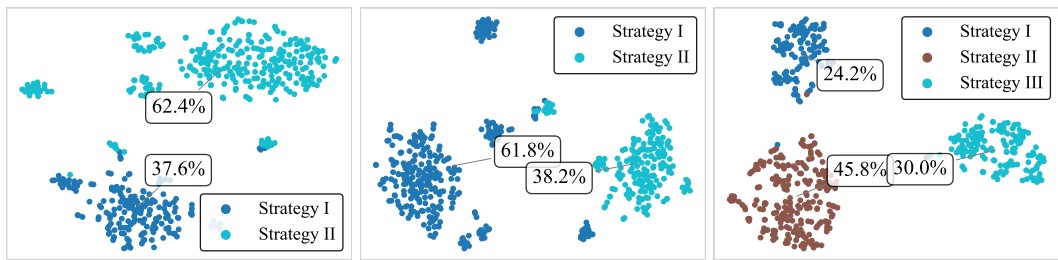

Figure 5: t-SNE visualization of embeddings (MiniLM encoder) for the 500 independent reasoning paths generated by GPT-4o-mini on three randomly selected AIME samples.

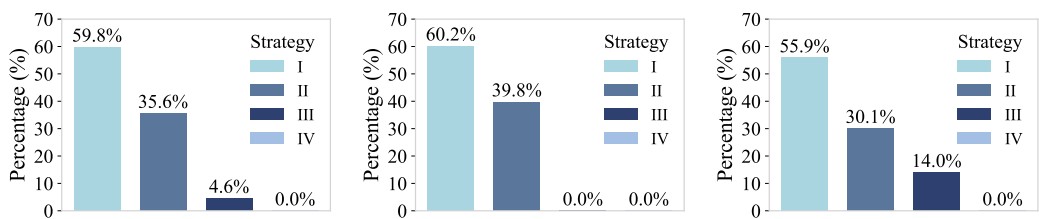

Figure 6: Percentage distribution of reasoning strategy clusters identified by OpenAI-o3 for the 500 independent reasoning paths generated by GPT-4o-mini on three randomly selected AIME samples.

ternatives. As our work shows, such bias plays a decisive role in shaping scaling effectiveness, highlighting the need for frameworks that explicitly model and balance strategy distributions.

### A.4 EXPERIMENTAL VALIDATION OF THEOREM 3.5

To validate Theorem 3.5, we conduct experiments on GPT-4o-mini using a set of samples selected from the AQuA dataset. For each sample, three scenarios are considered: low-complexity dominance, uniform sampling, and high-complexity dominance. In each scenario, 400 independent reasoning paths are generated: under low-complexity dominance, the low-complexity strategy receives 250 samples while the remaining strategies equally share the rest of the budget; under uniform sampling, all strategies evenly share the 400 samples; under high-complexity dominance, the high-complexity strategy receives 250 samples while the remaining strategies equally share the rest.

Performance is evaluated by the proportion of correct paths within the total budget. As shown in Figure 7, performance under low-complexity dominance exceeds that of uniform sampling, whereas high-complexity dominance leads to a substantial decline, thereby corroborating Theorem 3.5.

### A.5 THE RELATIONSHIP AMONG COMPLEXITY, ENTROPY, AND ACCURACY

To further understand the internal connections among strategy complexity, entropy, and accuracy, we conduct experiments on six representative samples from the AQuA dataset using GPT-4o-mini. For each problem, we coarsely identify (*i.e.,* Uniform-C (Z) method) four distinct solution strategies and generate 50 reasoning paths per strategy. We then employ a stronger LLM (OpenAI-o3) to estimate the minimum token requirement of each strategy, compute the entropy of the resulting answer distribution, and measure the proportion of correct answers.

As shown in Figure 8, the results consistently demonstrate a clear correlation among the three factors. Strategies with lower minimum token requirements generally correspond to lower entropy values, indicating more stable and confident predictions. In turn, these strategies also achieve higher accuracy, as reflected by the larger proportion of correct reasoning paths. Conversely, strategies with higher token requirements tend to exhibit higher entropy and reduced accuracy, suggesting that more complex reasoning processes are prone to unstable solution patterns and less reliable outcomes.

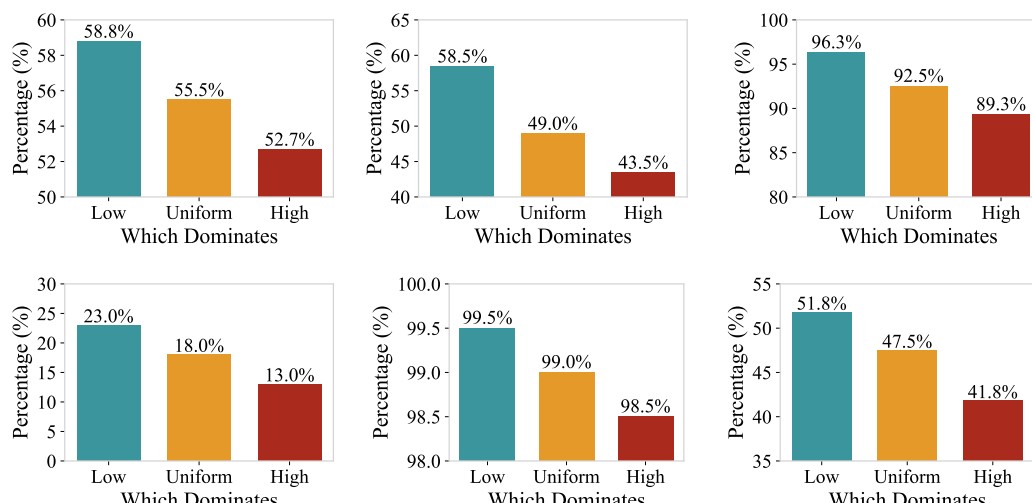

Figure 7: Experimental results on GPT-4o-mini with six AQuA samples under three scenarios (*i.e.,* low-complexity dominance, uniform sampling, and high-complexity dominance).

### A.6 Effectiveness of Filtering Strategies

We further examine the role of the filtering strategy, where $n$ denotes the number of strategies removed after ranking all strategies by their entropy. Specifically, the $n$ strategies with the highest entropy values are discarded. To evaluate this effect, we adopt Uniform-C (F) and conduct experiments on both GPT-4o-mini and GPT-4.1-mini. For each dataset, 16 strategies are generated under GPT-4o-mini and 8 strategies under GPT-4.1-mini, with four reasoning paths sampled per strategy.

The results, shown in Figure 9, exhibit consistent patterns across datasets. For GPT-4o-mini, performance improves when a moderate number of high-entropy strategies are removed, peaking at around 4–8 discarded strategies. However, overly aggressive filtering degrades performance by inadvertently eliminating some low-complexity strategies and reducing overall path diversity. In contrast, GPT-4.1-mini shows markedly lower sensitivity to $n$, with performance remaining relatively stable across a broader range of values. This indicates that stronger models produce more stable and reliable reasoning paths, making them inherently less dependent on entropy-based filtering.

Overall, these findings highlight that entropy-based filtering plays a more critical role in weaker reasoning models (*e.g.,* GPT-4o-mini), where eliminating unstable strategies substantially improves robustness. For stronger models (*e.g.,* GPT-4.1-mini), the framework is more stable and less sensitive to the choice of $n$, underscoring the general applicability of TTS-Uniform across different LLMs.

### A.7 Ideal v.s. Actual Token Consumption

To illustrate the discrepancy between theoretical reasoning and actual model generations, we compare the minimal token requirement with the token consumption of LLM-generated solutions on several mathematical problems. The minimal token solution represents the intrinsic complexity of a strategy, *i.e.,* the least number of steps required to derive the correct answer. In contrast, the actual generations often include redundant checks, explanatory phrases, and repeated steps, leading to much higher token usage. In particular, the theoretical minimal-token reasoning is obtained using the stronger OpenAI-o3 model through carefully designed prompting with multiple selections, whereas the actual LLM-generated reasoning is produced directly by GPT-4o-mini.

As shown in the two brown boxes below, the gap between the two is substantial. For example, in the marble problem, the minimal requirement is only 35 tokens, whereas the model output consumes 148 tokens. In the discount problem, the minimal solution requires 46 tokens, but the generated reasoning expands to 181 tokens. These results consistently reveal that actual token consumption in LLM reasoning far exceeds the theoretical minimum.

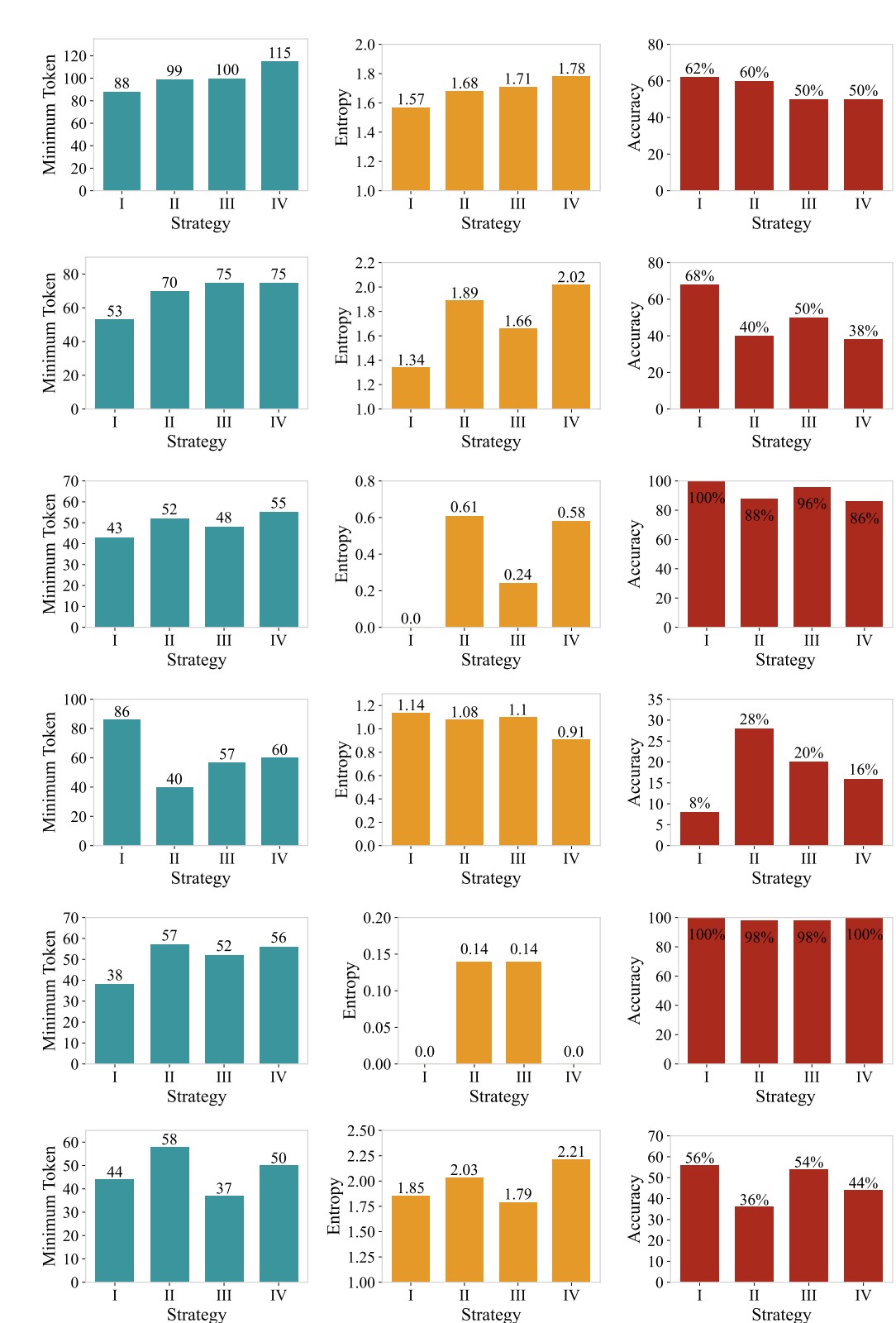

Figure 8: Strategy complexity, entropy, and accuracy across six AQuA samples.

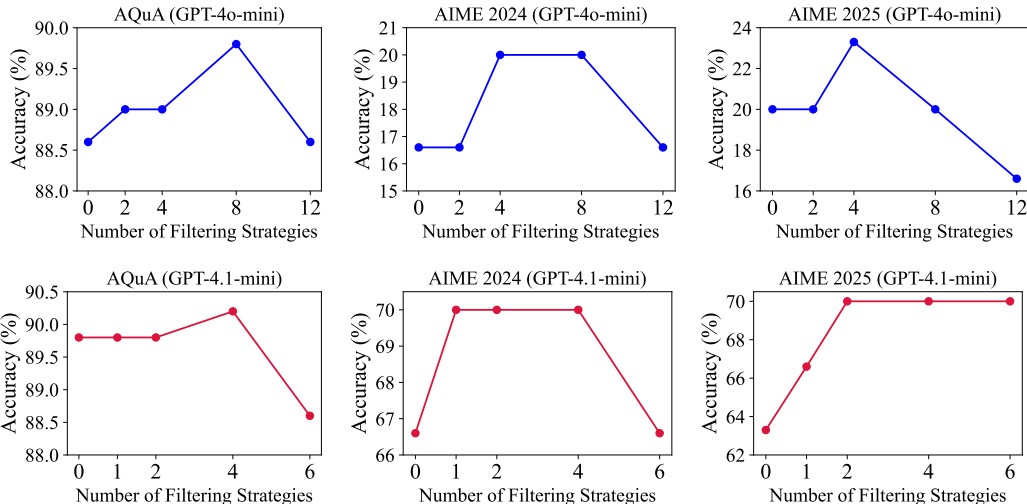

Figure 9: Effect of the filtering strategy number $n$ on the performance of Uniform-C (F), evaluated on GPT-4o-mini (top row) and GPT-4.1-mini (bottom row) across all datasets.

This observation directly supports our claim that using average token length of generated reasoning paths as a proxy for strategy complexity is unreliable. Since LLMs systematically produce outputs that deviate from the true minimum due to repetition and redundancy, average token length overestimates complexity and fails to capture the intrinsic difficulty of a strategy.

---

**Ideal v.s. Actual Token Consumption**

**Given Problem $Q$:** 20 marbles were pulled out of a bag of only white marbles, painted black, and then put back in. Then, another 20 marbles were pulled out, of which 1 was black, after which they were all returned to the bag. If the percentage of black marbles pulled out the second time represents their percentage in the bag, how many marbles in total Q does the bag currently hold? Answer Choices: (A) 40 (B) 200 (C) 380 (D) 400 (E) 3200.

*(Minimal Token Requirement)* Step 1: Bag has 20 black marbles. Probability black = $20/Q$. Step 2: In 20 draws, 1 black -> proportion = 1/20. Step 3: Equation: $20/Q$ = $1/20$. Solve $\Rightarrow$ $Q = 400$. Answer: $\boxed{400}$. **(Token Count = 35)**

*(Actual Token Usage)* Step 1: The bag initially has only white marbles, but 20 were painted black and returned, so there are exactly 20 black marbles in the bag, making the probability of drawing a black marble equal to $\frac{20}{Q}$, where $Q$ is the total number of marbles currently in the bag. Step 2: In a second round of drawing 20 marbles, exactly 1 black marble was observed, hence the observed proportion of black marbles in this draw is $\frac{1}{20}$, which is assumed to reflect the true proportion in the bag. Step 3: Equating the theoretical and observed probabilities gives the equation $\frac{20}{Q}$ = $\frac{1}{20}$; solving this equation by cross-multiplication yields $Q = 400$. Answer: The total number of marbles in the bag is $\boxed{400}$. **(Token Count = 148)**

---

### Ideal v.s. Actual Token Consumption

**Given Problem** $Q$: The original price of an item is discounted 22%. A customer buys the item at this discounted price using a \$20-off coupon. There is no tax on the item, and this was the only item the customer bought. If the customer paid \$1.90 more than half the original price of the item, what was the original price of the item? Answers: (A) \$61 (B) \$65 (C) \$67.40 (D) \$70 (E) \$78.20.

*(Minimal Token Requirement)* Step 1: Let original price $P$. Discounted price $= 0.78P$. Step 2: After coupon: $0.78P-20$. Step 3: Condition: $0.78P - 20 = \frac{1}{2}P + 1.90$. Step 4: Solve: $0.78P - 0.5P = 21.90$ $\Rightarrow 0.28P = 21.90 \Rightarrow P = 78.21$. Answer: $\boxed{78.20}$ (E). **(Token Count = 46)**

*(Actual Token Usage)* Step 1: Let the original price be denoted as $P$; applying a 22% discount results in a discounted price of $0.78P$, since $100\% - 22\% = 78\%$, and $78\% = 0.78$ in decimal form. Step 2: After applying the additional coupon worth \$20, the new effective price becomes $0.78P - 20$, subtracting the coupon value directly from the discounted price. Step 3: According to the condition provided in the problem, this final price equals half the original price plus an additional amount of \$1.90, so we set up the equation $0.78P - 20 = \frac{1}{2}P + 1.90$. Step 4: To isolate $P$, subtract $\frac{1}{2}P$ from both sides: $0.78P - \frac{1}{2}P - 20 = 1.90$, which simplifies to $0.28P = 21.90$, and solving this gives $P = \frac{21.90}{0.28} = 78.21$. Answer: Rounding to the nearest cent as needed, the original price is approximately $\boxed{78.20}$, corresponding to option (E). **(Token Count = 181)**

---

**Example Problem A: Different Reasoning Strategies**

**Given Problem $\mathcal{Q}$.** Find all real $c$ such that the line $y = 2x + c$ intersects the parabola $y = x^2$ at two points $A, B$ with $AB = \sqrt{20}$.

*(Algebraic Strategy $\mathcal{R}_1$)*

- **Step 1 (Intersections).** Solving $x^2 = 2x + c$ gives the quadratic

$$x^2 - 2x - c = 0,$$

   with roots $x_1, x_2$. Two distinct intersection points require the discriminant $\Delta = 4 + 4c > 0$, *i.e., $c > -1$.*

- **Step 2 (Vieta).**
$$x_1 + x_2 = 2, \qquad x_1 x_2 = -c.$$

- **Step 3 (Chord length).** The points are $A = (x_1, x_1^2)$, $B = (x_2, x_2^2)$. Hence

$$AB^2 = (x_2 - x_1)^2 + (x_2^2 - x_1^2)^2 = (x_2 - x_1)^2 \Big[1 + (x_1 + x_2)^2\Big].$$

   Using $(x_2 - x_1)^2 = (x_1 + x_2)^2 - 4x_1 x_2$ and Vieta,

$$(x_2 - x_1)^2 = (2)^2 - 4(-c) = 4 + 4c, \qquad 1 + (x_1 + x_2)^2 = 1 + 4 = 5,$$

   so
$$AB^2 = (4 + 4c) \cdot 5 = 20(1 + c).$$

- **Step 4 (Impose $AB = \sqrt{20}$).**
$$AB^2 = 20 \implies 20(1 + c) = 20 \implies c = 0.$$

   This satisfies $c > -1$, so it is valid. For $c = 0$, the intersections are $x = 0, 2$, *i.e., $A = (0,0)$, $B = (2,4)$*, and indeed

$$AB = \sqrt{(2-0)^2 + (4-0)^2} = \sqrt{4 + 16} = \sqrt{20}.$$

*(Geometric Strategy $\mathcal{R}_2$)*

- **Step 1 (Symmetry parametrization).** Any chord of $y = x^2$ with slope 2 has endpoints symmetric about $x = 1$. Let

$$A = (1 - t, (1 - t)^2), \qquad B = (1 + t, (1 + t)^2) \quad (t > 0).$$

   Then
$$\text{slope}(AB) = \frac{(1 + t)^2 - (1 - t)^2}{(1 + t) - (1 - t)} = \frac{4t}{2t} = 2.$$

- **Step 2 (Chord length).**
$$AB^2 = \big((1 + t) - (1 - t)\big)^2 + \big((1 + t)^2 - (1 - t)^2\big)^2 = (2t)^2 + (4t)^2 = 20t^2.$$

   Set $AB = \sqrt{20} \Rightarrow 20t^2 = 20 \Rightarrow t = 1$, hence the endpoints are

$$A = (0, 0), \qquad B = (2, 4).$$

- **Step 3 (Recover $c$).** Since the line through $A, B$ has slope 2, it is $y = 2x + c$. Plugging $A = (0, 0)$ gives $0 = 0 + c \Rightarrow c = 0$.

**Conclusion.** The unique value is $\boxed{c = 0}$. (Note: the "two points" condition requires $c > -1$; $c = 0$ satisfies this and yields the desired chord length.)

---

### Example Problem B: Different Reasoning Strategies

**Given Problem** $\mathcal{Q}$. Evaluate

$$S_n = \sum_{k=0}^{n} k \binom{n}{k} \quad \text{for } n \geq 1.$$

***(Combinatorial Strategy $\mathcal{R}_1$)***

- **Step 1 (Combinatorial model).** Consider all pairs $(S, i)$ where $S \subseteq [n]$ is a subset and $i \in S$ is a distinguished (marked) element.

- **Step 2 (Count in two ways).**

  1. *First way (by subset size).* Fix $k = |S|$. Choose $S$ in $\binom{n}{k}$ ways, then choose the marked element $i$ inside $S$ in $k$ ways. Total over all $k$:

  $$\#\{(S, i)\} = \sum_{k=0}^{n} k \binom{n}{k} = S_n.$$

  2. *Second way (by the marked element first).* Choose the marked element $i$ first: $n$ choices. The remaining elements $[n] \setminus \{i\}$ may be included or not independently, giving $2^{n-1}$ choices for $S$ containing $i$. Thus

  $$\#\{(S, i)\} = n \cdot 2^{n-1}.$$

- **Step 3 (Equate the counts).** Therefore $S_n = n\, 2^{n-1}$.

***(Generating-Function Strategy $\mathcal{R}_2$)***

- Using $(1 + x)^n = \sum_{k=0}^{n} \binom{n}{k} x^k$, differentiate:

$$n(1 + x)^{n-1} = \sum_{k=0}^{n} k \binom{n}{k} x^{k-1}.$$

  Multiply both sides by $x$ and set $x = 1$:

$$\sum_{k=0}^{n} k \binom{n}{k} = n \cdot 1 \cdot (1 + 1)^{n-1} = n\, 2^{n-1}.$$

**Conclusion.** $\boxed{\sum_{k=0}^{n} k \binom{n}{k} = n\, 2^{n-1}}$.

---

### Example A (Coarse-Grained): Algebraic vs Geometric

**Given Problem $\mathcal{Q}$.** Find all real $c$ such that the line $y = 2x + c$ meets the parabola $y = x^2$ at two points $A, B$ with $|AB| = \sqrt{20}$.

**Strategy set** $\mathcal{S} = \{\mathcal{R}_{\text{alg}}, \mathcal{R}_{\text{geo}}\}$.

*($\mathcal{R}_{alg}$ — Algebraic (Vieta + identities))*

1. Intersections satisfy $x^2 = 2x + c \Rightarrow x^2 - 2x - c = 0$ with roots $x_1, x_2$.

2. $A(x_1, x_1^2), B(x_2, x_2^2)$. Then
$$|AB|^2 = (x_2 - x_1)^2 + (x_2^2 - x_1^2)^2 = (x_2 - x_1)^2 \left[1 + (x_1 + x_2)^2\right].$$
   By Vieta, $x_1 + x_2 = 2 \Rightarrow |AB|^2 = 5(x_2 - x_1)^2$.

3. Given $|AB|^2 = 20 \Rightarrow (x_2 - x_1)^2 = 4$. For $x^2 - 2x - c = 0$, $(x_2 - x_1)^2 = 4 + 4c$. Hence $4 + 4c = 4 \Rightarrow c = 0$.

*($\mathcal{R}_{geo}$ — Geometric/parametric (completed square))*

1. $(x - 1)^2 = 1 + c \Rightarrow x = 1 \pm \sqrt{1 + c}$.

2. $\Delta x = 2\sqrt{1 + c}, \Delta y = (x_2^2 - x_1^2) = (x_2 - x_1)(x_2 + x_1) = 2\sqrt{1 + c} \cdot 2 = 4\sqrt{1 + c}$.

3. $|AB| = \sqrt{(2\sqrt{1 + c})^2 + (4\sqrt{1 + c})^2} = \sqrt{20(1 + c)}$. Set $|AB| = \sqrt{20} \Rightarrow 1 + c = 1 \Rightarrow c = 0$.

**Equivalence annotation:** This example naturally forms *coarse-grained* classes: $\{\mathcal{R}_{\text{alg}}\}$ (algebraic) vs. $\{\mathcal{R}_{\text{geo}}\}$ (geometric). Local step choices within each class (e.g., discriminant vs. Vieta; parameterization variants) are considered equivalent under the same high-level reasoning logic.

**Answer.** $c = 0$.

---

### Example B (Fine-Grained): Two Step-Level Variants of Radical Elimination

**Given Problem $\mathcal{Q}$.** Solve
$$\sqrt{x + 9} + \sqrt{25 - x} = 8, \qquad x \in [-9, 25].$$

**Coarse view.** Both workable methods belong to the same high-level strategy: "algebraic elimination of radicals." Coarse-grained equivalence would merge them.

**Fine-grained strategies** $\mathcal{S} = \{\mathcal{R}_{\text{sq-expand}}, \mathcal{R}_{\text{isolate}}\}$.

*($\mathcal{R}_{sq\text{-}expand}$ — Square the sum, then square again)*

1. Square both sides: $(\sqrt{x + 9} + \sqrt{25 - x})^2 = 64 \Rightarrow x + 9 + 25 - x + 2\sqrt{(x + 9)(25 - x)} = 64$. Thus $\sqrt{(x + 9)(25 - x)} = 15$.

2. Square again: $(x + 9)(25 - x) = 225 \Rightarrow -x^2 + 16x = 0 \Rightarrow x \in \{0, 16\}$.

3. Check in the original equation: both $x = 0, 16$ satisfy it.

*($\mathcal{R}_{isolate}$ — Isolate one radical, square twice)*

1. $\sqrt{x + 9} = 8 - \sqrt{25 - x}$ (RHS nonnegative holds on the domain).

2. Square: $x + 9 = 64 - 16\sqrt{25 - x} + 25 - x \Rightarrow 2x - 80 = -16\sqrt{25 - x}$.

3. Hence $\sqrt{25 - x} = (40 - x)/8$. Square again: $25 - x = (40 - x)^2/64 \Rightarrow x(x - 16) = 0 \Rightarrow x \in \{0, 16\}$.

4. Check: both $x = 0, 16$ are valid.

**Equivalence annotation:** Under *fine-grained equivalence*, $\mathcal{R}_{\text{sq-expand}}$ and $\mathcal{R}_{\text{isolate}}$ are distinct due to the concrete step sequences (square-sum-first vs. isolate-then-square), despite sharing the same coarse algebraic nature.

**Answer.** $x \in \{0, 16\}$.

