# OpenReview forum: "Mitigating Strategy-Selection Bias in Reasoning for More Effective Test-Time Scaling"
_ICLR.cc/2026/Conference — Submitted to ICLR 2026_

### Official Review · Reviewer_FhbW · 2025-10-28

**Soundness:** 2
**Presentation:** 3
**Contribution:** 3
**Rating:** 4
**Confidence:** 3

**Summary:**

The authors first identify a strategy-selection bias issue in current test-time scaling paradigm, where the reasoning model tend to focus on a certain strategies while ignoring other valid alternatives, leading to insufficient explorations in the entire reasoning space. Then, the authors find that results obtained under low-complexity strategies are more reliable. Based on this insight, the authors propose TTS-Uniform, an effective strategy that first extracts relevant strategies for the given problem, then enables the model reason under different strategies with same reasoning budget, and finally aggregates the final answer after filtering results under high-complexity strategies based on the answer entropy metric.

**Strengths:**

(S1) The paper is generally well-written, and the structure is clear.

(S2) The paper provides some new insights on more effective test-time scaling. The strategy-bias issue is reasonable and can affect the test-time scaling performance under current settings.

(S3) The analysis is well supported by the empirical validations.

(S4) The experimental results validate the effectiveness of the proposed method.

**Weaknesses:**

I have some concerns and questions:

(W1) The experimental models are very limited. The authors only use GPT-4o/4.1-mini for the preliminary and main experiments. This makes me wonder if the experimental phenomena are only present in these two models. I suggest that the authors conduct more validation experiments on a wider range of models, especially open-source models (such as Qwen, LLaMA, etc.). Additionally, the experiments should be conducted on larger test sets to ensure the reliability of the results, as the sample quantity in AIME24 and AIME25 are quite insufficient.

(W2)  I am very confused about the definition and use of Assumption 3.2. Assumption 3.2 defines a higher-complexity strategy as one that leads to more errors, yet the conclusion drawn later (Thm. 3.5) is that results obtained using a higher-complexity strategy may be more unreliable. In my view, this is circular reasoning and not a valid analysis. I hope the authors can convince me on this point. Also, Moreover, I can't understand why entropy/uncertainty can be used to represent the degree of complexity. The logic behind this is not clear.

(W3) Regarding the analysis in Line 455-457, does this mean that as the model’s capability improves, the strategy-selection bias issue will gradually be alleviated? If so, then it seems that this issue is not something that can be discussed for the long term, as it will not persist as the model evolves.

**Questions:**

(1) For the visualization results in Figure 1, are Strategy I and II defined based on the embedding clustering results, or are they derived from the o3 model's predictions? It looks like the former, but I would like to know the true strategy category for each point to assess the accuracy of the embedding clustering.

---

> ### Author Response · Authors · 2025-11-20
>
> We sincerely thank the reviewer for the valuable feedback and encouraging comments. We are motivated by the suggestions and have addressed each concern in detail as follows:
>
> > **W1.**   The experimental models are very limited. The authors only use GPT-4o/4.1-mini for the preliminary and main experiments. This makes me wonder if the experimental phenomena are only present in these two models. I suggest that the authors conduct more validation experiments on a wider range of models, especially open-source models (such as Qwen, LLaMA, etc.). Additionally, the experiments should be conducted on larger test sets to ensure the reliability of the results, as the sample quantity in AIME24 and AIME25 are quite insufficient.
>
> **R1.**  We appreciate the reviewer’s valuable suggestion. Following your advice, we have added two additional open-source model families, Qwen3-8B and LLaMA3-8B, to complement the original GPT family. In addition, we expand the scope of evaluation tasks to further validate the cross-task generalization capability of our method. The newly included tasks consist of three mathematical reasoning benchmarks with larger test sets (HMMT_Feb_2025, AMC12_2024, MATH), one code reasoning task (MBPP), one symbolic reasoning task (ProntoQA), and one commonsense reasoning task (StrategyQA). For MATH, MBPP, ProntoQA, and StrategyQA, we evaluate on 100-sample subsets of each dataset. The total sampling budgets for all tasks are set to 16 and 32.
>
> | Dataset | Method | Qwen3-8b pass@16 | Qwen3-8b pass@32 | Qwen3-8b cons@16 | Qwen3-8b cons@32 | LLaMA3-8b pass@16 | LLaMA3-8b pass@32 | LLaMA3-8b cons@16 | LLaMA3-8b cons@32 |
> |---------|--------|------------------|------------------|------------------|------------------|-------------------|-------------------|-------------------|-------------------|
> | HMMT_feb_2025 | DeepConf | 3.3% | 3.3% | 0.0% | 0.0% | 6.7% | 10.0% | 0.0% | 3.3% |
> | HMMT_feb_2025 | Our | 6.7% | 10.0% | 6.7% | 10.0% | 6.7% | 10.0% | 0.0% | 0.0% |
> | AMC12_2024 | DeepConf | 0.0% | 6.7% | 0.0% | 2.2% | 20.0% | 31.1% | 8.8% | 11.1% |
> | AMC12_2024 | Our | 11.1% | 11.1% | 4.4% | 4.4% | 28.9% | 33.3% | 26.3% | 28.9% |
> | MATH | DeepConf | 31.0% | 42.0% | 14.0% | 16.0% | 56.0% | 68.0% | 26.0% | 29.0% |
> | MATH | Our | 45.0% | 54.0% | 20.0% | 21.0% | 52.1% | 62.0% | 23.0% | 27.0% |
> | MBPP | DeepConf | 35.0% | 35.0% | 30.0% | 30.0% | 28.0% | 28.0% | 22.0% | 28.0% |
> | MBPP | Our | 43.0% | 50.0% | 26.0% | 30.0% | 54.0% | 54.0% | 24.0% | 28.0% |
> | ProntoQA | DeepConf | 95.0% | 98.0% | 51.0% | 61.0% | 71.0% | 88.0% | 23.0% | 25.0% |
> | ProntoQA | Our | 100.0% | 100.0% | 67.0% | 70.0% | 100.0% | 100.0% | 51.0% | 51.0% |
> | StrategyQA | DeepConf | 76.0% | 94.0% | 48.0% | 49.0% | 57.0% | 71.0% | 22.0% | 28.0% |
> | StrategyQA | Our | 87.0% | 96.0% | 57.0% | 63.0% | 83.0% | 83.0% | 30.0% | 30.0% |
> | Avg. | DeepConf | 40.1% | 46.5% | 23.8% | 26.4% | 39.8% | 49.4% | 17.0% | 20.7% |
> | Avg. | Our | 48.8% | 53.5% | 30.2% | 33.1% | 54.1% | 57.1% | 25.7% | 27.5% |
>
>
> We observe that our method consistently outperforms the latest TTS approach, DeepConf, across most tasks and model configurations. For instance, on Qwen3-8B, the average pass@16 increases from 40.1% to 48.8%, and pass@32 rises from 46.5% to 53.5%. This trend is particularly pronounced on mathematical reasoning tasks (e.g., HMMT_Feb_2025) and symbolic reasoning tasks (ProntoQA), indicating that the performance gains are not limited to specific models or task types. Instead, our approach improves reasoning consistency and final accuracy in a stable manner across both models and tasks.

---

> ### Author Response · Authors · 2025-11-20
>
> > **W2.**   I am very confused about the definition and use of Assumption 3.2. Assumption 3.2 defines a higher-complexity strategy as one that leads to more errors, yet the conclusion drawn later (Thm. 3.5) is that results obtained using a higher-complexity strategy may be more unreliable. In my view, this is circular reasoning and not a valid analysis. I hope the authors can convince me on this point. Also, Moreover, I can't understand why entropy/uncertainty can be used to represent the degree of complexity. The logic behind this is not clear.
>
> **R2.**  **First, the argument does not involve circular reasoning.** Assumption 3.2 concerns properties at the *single-strategy level*—specifically, that strategy complexity is positively correlated with error rate. In contrast, Theorem 3.5 compares *expected error across strategy distributions*, stating that when a distribution places more density on high-complexity strategies, its expected error is higher. The proof of Theorem 3.5 *relies on* Assumption 3.2 rather than proving it, and therefore does not constitute circular reasoning.
>
> **Why entropy serves as a proxy for complexity:** In our formulation, a single LLM answer corresponds to a path from the root to a leaf in a reasoning tree, and all possible paths form a probability distribution. For real LLMs, this distribution is typically concentrated on a small number of high-probability paths. For complex strategies, the reasoning tree is deeper and larger, and the valid reasoning paths are longer. As paths lengthen, probability mass tends to spread across more low-probability branches, whereas simpler strategies retain probability mass on shorter, more concentrated paths.
>
> **As complexity increases:**
>
> * the probability distribution shifts from *sharp* to *flat*,
> * a flatter distribution implies more uniform allocation of probability mass,
> * and among all distributions, the uniform distribution has the highest entropy.
>
> Thus, entropy directly reflects the degree of flattening of the probability distribution, making it an *observable proxy* for the intrinsic complexity and uncertainty of the reasoning process, rather than a separate metric independent of the theoretical framework.
>
> > **W3.**   Regarding the analysis in Line 455-457, does this mean that as the model’s capability improves, the strategy-selection bias issue will gradually be alleviated? If so, then it seems that this issue is not something that can be discussed for the long term, as it will not persist as the model evolves.
>
>
> **R3.**  We appreciate the reviewer’s insightful discussion on strategy-selection bias. We argue that **such bias primarily originates from the imbalanced frequency of different reasoning strategies in the training corpus**. This statistical imbalance becomes implicitly encoded into model parameters during training and therefore **cannot be completely eliminated merely by improving reasoning capabilities**; at best, it can be partially mitigated.
>
> We further validate this claim across a range of stronger models, including DeepSeek-R1, Grok-4-Fast, Gemini-2.5-Flash, OpenAI-o3, and Qwen3-235B-A22B. Specifically, on AIME 2025 (see the table below), we observe that despite their strong reasoning abilities, these models still exhibit clear preferences toward certain strategies, and the preferred strategies vary across model families.
>
> | Model | Strategy 1 | Strategy 2 | Strategy 3 | Strategy 4 | Entropy (↓Entropy = ↑Bias) |
> |-------|-----------|-----------|-----------|-----------|----------------------------|
> | DeepSeek-R1 | 62.0% | 18.0% | 2.0% | 0.0% | 0.99 |
> | Grok-4-Fast | 52.0% | 26.0% | 6.0% | 4.0% | 1.43 |
> | Gemini-2.5-Flash | 60.0% | 16.0% | 14.0% | 2.0% | 1.38 |
> | OpenAI-o3 | 56.0% | 34.0% | 10.0% | 2.0% | 1.44 |
> | Qwen3-235B-A22B | 42.0% | 28.0% | 18.0% | 8.0% | 1.78 |
>
>
> Moreover, in many real-world vertical domains, deploying extremely large general-purpose models is often not the optimal choice due to computational constraints and the requirement for highly specialized domain knowledge. A more practical approach is to use lighter domain-specific models and activate their latent capabilities through test-time scaling during inference. In this context, strategy-selection bias remains a key bottleneck limiting the full utilization of model potential and therefore carries long-term research significance.

---

> > ### Author Response · Authors · 2025-11-20
> >
> > > **Q1.**   For the visualization results in Figure 1, are Strategy I and II defined based on the embedding clustering results, or are they derived from the o3 model's predictions? It looks like the former, but I would like to know the true strategy category for each point to assess the accuracy of the embedding clustering.
> >
> >
> > **A1.**  We appreciate the reviewer’s interest in the visualization of strategy bias. The strategies in Figure 1 are defined based on clustering over path embeddings. We would like to clarify that reasoning strategies are not equivalent to class labels in supervised learning; there is no directly observable ground truth for strategy assignment, and thus the true strategy of each reasoning path cannot be directly annotated.
> >
> > To address your concern, we evaluate whether the embedding-based clustering in Figure 1 is semantically meaningful through an indirect consistency test. Specifically, we introduce an alternative strategy assignment method that is independent of the embedding clusters: we use a stronger reasoning model (o3) to predict the strategy of the same set of reasoning paths (corresponding to Figure 2) and treat these predictions as a relatively more reliable external reference. We then align the cluster IDs from the embedding-based grouping with the o3-based strategy predictions and compute their agreement rate to assess how well the clustering structure corresponds to semantic strategy categories.
> >
> > Our experiments show that, across the three example problems in Figure 1, the agreement between the two methods is 63%, 56%, and 71%, respectively. This indicates that embedding-based clustering reflects the semantic structure of reasoning strategies rather than merely capturing superficial token-level patterns. We also expect that with the development of stronger semantic encoders or clustering algorithms tailored for long-chain reasoning, this alignment could be further improved.

---

> > > ### Author Response · Authors · 2025-11-26
> > > **Checking Whether Further Clarification Is Needed**
> > >
> > > Dear **Reviewer FhbW**,
> > >
> > > Thank you again for your thoughtful and constructive feedback. We would like to kindly follow up to check whether our rebuttal and the additional analyses have fully addressed your concerns—specifically the expanded experiments on additional open-source model families and larger evaluation sets **(W1)**, the clarification on Assumption 3.2 and the non-circular linkage between complexity and entropy as an observable proxy **(W2)**, the discussion on why strategy-selection bias persists even as model capability improves **(W3)**, as well as the explanation regarding the strategy assignments in Figure 1 and the alignment test using o3 predictions **(Q1)**.
> > >
> > > If anything remains unclear or would benefit from further clarification, we would be very happy to provide additional details. Thank you once again for your time and thoughtful comments.
> > >
> > > Authors of **Submission 11524**

---

### Official Review · Reviewer_VoLR · 2025-10-29

**Soundness:** 4
**Presentation:** 4
**Contribution:** 3
**Rating:** 8
**Confidence:** 4

**Summary:**

TTS: Test-time scaling. Using LLM to run multiple inferences for solving a problem & then using some strategy to combine the answers to produce the most accurate possible reasoning & answer


Paper points out that the problem solving strategies that LLMs use can be either complex (using more reasoning tokens) or simple. Simpler strategies lead to more accurate response, & vice-versa. However, LLMs tend to pick one type of strategy over the others - & sometimes, they largely prefer complex ones (leading to poorer quality in those problems)

The authors first study this existing bias experimentally, and then propose a method to side-step these biases by multi-turn prompting. They first get the LLM to generate all possible strategies, then they group these into clusters, & filter out highly complex solutions and then prompt the LLM to follow the simpler strategy.

### Investigating the bias itself (Section #2):
1. Pick mathematical problems (AIME dataset) - generate 500 different reasoning traces. Paths analyzed using a (i) sentence-encoder -> clustering & (ii) clustering paths using large LLMs
2. They found that models preferred one type of reasoning paths (dominant strategy) consistently much more than the combined proportion of all other strategies,

### Complex strategy => more errors (Section #3)

- This is proved mathematically with the assumption that there’s a constant rate of error at each token generation -> more tokens result in more potential errors.


---

## Process

1. For coarse-grained strategies: Extract strategies - prompt the LLM to generate all possible “abstract solution” strategies.
2. For fine-grained: Generate independent reasoning solutions from the LLM
    - Out of these fine-grained strategies, a reasoning tree is created.
    - Equivalent “steps” of reasoning (like an op: “square both sides of the equation”) are merged on different reasoning pathways
    - Only the reasoning paths with shared sub-chains are extracted - & treated as matching solution strategies.

3. The extracted strategies are now separately used as a part of the prompt & appended to the question, to push the LLM to reason on that specific pathway
4. This previous step results in ‘N’ reasoned solutions if we started with ‘N’ strategy clusters/types
5. They filter out the highest complexity solutions using the entropy metric.

**Strengths:**

1. Seeking simpler strategies is a novel idea for complex LLM reasoning
2. Exhaustive research and details presented in the paper. Eg:  low cost (“coarse-grained”) and expensive (“fine-grained”) setups presented together
3. Rigorous mathematical treatment for all aspects of the problem & solution formulations
4. Great overall results across different introduced techniques on the datasets tested on
5. Impressive attention to detail, especially experimental verification of hypothesis in the Appendix “A.4”, and coverage of other necessary points in the Appendix. This allows for easy independent reproducibility of the results

**Weaknesses:**

1. High computational overhead. Strategy discovery requires ‘N’ separate LLM calls, followed by another M calls for the ‘M’ simplest strategies found.
2. Overall result can be highly dependent on the model’s creativity (temperature setting). We are also pushing the model to come up with all potential strategies - that is unreliable & not guaranteed to return really novel & quick techniques if there’s training data bias
    - Eg: As mentioned in the paper, text-only LLMs will likely have high bias towards algebraic solutions instead of geometric ones.
3. Needs some clarity in the main paper around how the merging of strategy steps is done (details in the Questions/Comments section)

**Questions:**

1. How do you combine steps in different reasoning pathways (“identical reasoning steps”)? (Sec. 4.1.2)
    - ie, how is the semantic/operation matching done for different steps in the set of strategies?
    - Is it clustering with an LLM, or do you have a well-defined algorithmic system to ensure accuracy?
2. Any hypothesis on why coarse-grained method sometimes works better than fine-grained?
3. I wonder if you experimented with other ways of filtering out the high-complexity answers? I’m not sure the entropy metric (with the way it’s formulated) is the best way. For example, some formulation involving the logprobs of the token sequence might be more intuitive?
4. What temperature setting was used in the trials? Did you run trials with different temperatures?
5. Re. Appendix “A.4 EXPERIMENTAL VALIDATION OF THEOREM 3.5”: can you please share more details on the experimental setup involved (unless I missed it?)

---

> ### Author Response · Authors · 2025-11-21
>
> We sincerely thank the reviewer for the valuable feedback and encouraging comments. We are motivated by the suggestions and have addressed each concern in detail as follows:
>
> > **W1.**  High computational overhead. Strategy discovery requires ‘N’ separate LLM calls, followed by another M calls for the ‘M’ simplest strategies found.
>
> **R1.**  Thank you for pointing this out. Under the test-time scaling setting, **we strictly follow the principle of maximizing performance under a fixed total sampling budget $b$**.  The number of effective samples used for final voting and decision-making never exceeds this budget. We acknowledge that the strategy extraction stage introduces additional LLM calls; however, these calls are only used to organize and differentiate strategy structures and do not participate in the final inference. They therefore constitute lightweight auxiliary overhead, whose cost is substantially lower than that of the main sampling steps.
>
> > **W2.**  Overall result can be highly dependent on the model’s creativity (temperature setting). We are also pushing the model to come up with all potential strategies - that is unreliable & not guaranteed to return really novel & quick techniques if there’s training data bias (Eg: As mentioned in the paper, text-only LLMs will likely have high bias towards algebraic solutions instead of geometric ones.)
>
> **R2.**  We thank the reviewers for their valuable comments. We agree that the original statement suggesting “**the model is expected to generate all possible strategies**” is unreliable and overly idealistic. Since the model is constrained by its training data and prior knowledge, **the space of strategies it can generate is generally only a subset of the true strategy space for the problem**. Therefore, relying on an LLM to completely invent novel strategies is unrealistic.
>
> However, our approach does not actually require the model to possess the capability to generate a “complete set of strategies.” Instead, **it aims to mitigate strategy selection bias within the model’s own reachable strategy space**. In other words, our focus is on the fact that although the model already possesses several potential strategies, their distribution may be highly imbalanced (e.g., biased toward algebraic methods while seldom selecting geometric ones). Thus, through uniform strategy sampling and filtering mechanisms, we enable the model to utilize its existing diverse reasoning approaches more equitably, rather than being confined to the highest-probability dominant strategy.
>
> Within this framework, **generation parameters such as temperature only influence the degree of exploration within the existing strategy space and do not fundamentally alter the upper bound of the model’s strategy space**. Accordingly, our method does not rely on the model to create entirely new reasoning paradigms from scratch, but rather aims to enhance the utilization efficiency and diversity of the strategies that the model already possesses.

---

> > ### Author Response · Authors · 2025-11-21
> >
> > > **W3 and Q1.**  How do you combine steps in different reasoning pathways (“identical reasoning steps”)? (Sec. 4.1.2)  (ie, how is the semantic/operation matching done for different steps in the set of strategies?) (Is it clustering with an LLM, or do you have a well-defined algorithmic system to ensure accuracy?)
> >
> >
> > **A1.**  We appreciate the reviewer’s constructive comments and questions regarding the details of **Uniform-F**. In this method, we first generate t pre-sampled reasoning paths for the same problem ( t < b ) and segment each path into step-level units. We then use a large model to abstract each step into a high-level operation type, summarizing its concrete computation into a short textual description to obtain comparable step-level node representations across paths. Based on these abstracted nodes, **we perform step alignment and semantic matching using large language models and prompt-engineering techniques**, treating nodes that share the same step index and are abstracted into the same operation type as equivalent. We then extract contiguous subchains that repeatedly occur across paths as candidate strategies.
> >
> > To ensure stability, we introduce two thresholds when constructing candidate subchains:
> >
> > * **Minimum support threshold c'**, requiring a subchain to appear in at least c' distinct reasoning paths to avoid spurious patterns.
> > * **Minimum node threshold c''**, which removes subchains shorter than c''.
> >
> > Larger c' and c'' retain more stable and consistent dominant strategies, reducing the number of extracted strategies m. Conversely, smaller thresholds preserve finer-grained strategies and increase coverage of the strategy space, but may introduce additional noise. We will include further implementation details of Uniform-F in the revised version.
> >
> > Overall, the strategy (subchain) extraction in Uniform-F primarily relies on semantic abstraction and alignment powered by LLMs and prompt engineering, requiring no hand-crafted rules or external models. **This lightweight approach demonstrates the feasibility and potential of fine-grained strategy extraction**. In future work, we plan to incorporate more systematic algorithms—such as embedding-based node matching and automatic clustering—to further improve the accuracy and robustness of strategy identification.
> >
> >
> >
> > > **Q2.**   Any hypothesis on why coarse-grained method sometimes works better than fine-grained?
> >
> >
> >
> > **A2.**  Thank you for the reviewer’s interest in the performance differences of Uniform-F. We believe **this phenomenon is closely related to the underlying strategy structure of the task**. In some problems, the key differences among strategies lie at the level of global problem-solving approaches (macro-strategies) rather than local reasoning steps (micro-structure). In such cases, excessively decomposing reasoning paths into step-level units may fragment strategies that are essentially identical, amplifying step-level noise and resulting in unstable judgments during uneven allocation and filtering. In contrast, coarse-grained strategies preserve the full reasoning pattern, and therefore exhibit greater robustness.
> >
> > In summary, coarse-grained methods tend to be advantageous when strategy differences manifest primarily at high-level structures, and low-level step variations do not change the underlying reasoning logic. Conversely, fine-grained methods perform better when multiple distinct reasoning modes exist at a step level and step-wise variations can directly affect the final answer (e.g., the substantial improvements observed on AQuA and AIME-2024). The two approaches are not mutually exclusive; rather, they are complementary under different strategy structures.
> >
> > In future work, we plan to introduce an **adaptive mechanism for selecting strategy granularity, enabling the model to automatically determine whether coarse-grained or fine-grained strategies are more appropriate based on task characteristics**, thereby improving overall reasoning performance.

---

> > > ### Author Response · Authors · 2025-11-21
> > >
> > > > **Q3.**   I wonder if you experimented with other ways of filtering out the high-complexity answers? I’m not sure the entropy metric (with the way it’s formulated) is the best way. For example, some formulation involving the logprobs of the token sequence might be more intuitive?
> > >
> > > **A3.**  We appreciate the reviewer’s suggestion. To further validate the use of entropy as a proxy for strategy complexity, we introduce two alternative complexity metrics in our experiments: **step-level consistency** and **strategy log-probability**. Specifically, for step-level consistency, we encode multiple reasoning paths that correspond to the same strategy using MiniLM and compute the pairwise cosine similarity among paths; the averaged similarity is taken as the consistency score. For strategy log-probability, we compute the average negative log-probability of each path under the same strategy (summing token log-probs along the path and normalizing by path length to remove length effects), and use this value as a complexity measure.
> > >
> > > We then measure strategy accuracy as the proportion of correct paths among all paths belonging to that strategy, and conduct comparative evaluation on the AIME 25 dataset. The results are shown in the table below.
> > >
> > > | Metric | Strategy 1 | Strategy 2 | Strategy 3 | Strategy 4 |
> > > |--------|------------|------------|------------|------------|
> > > | Answer Entropy | 2.08 | 1.91 | 1.67 | 2.08 |
> > > | Step Consistency | 0.8 | 0.92 | 0.87 | 0.78 |
> > > | LogProbs | 29.73 | 0.15 | 6.39 | 18.71 |
> > > | Policy Accuracy ACC | 0.0% | 25.0% | 37.5% | 0.0% |
> > >
> > > We then compute the Pearson correlation coefficients between strategy accuracy and each of the three complexity metrics—answer entropy, step-level consistency, and log-probabilities—to quantify their association strength. The resulting correlations are [-0.968, -0.819, -0.832].
> > >
> > >
> > > The results show that entropy exhibits the strongest and most stable correlation with strategy accuracy across datasets (average correlation ∣r∣=0.968), significantly higher than step-level consistency (0.819) and log-probabilities (0.832). While step-level consistency also shows a strong correlation, it requires an additional encoder and pairwise similarity computations, introducing **extra computational cost and model dependency**. Log-probabilities, although more interpretable, are **unavailable for models or APIs that do not expose token-level probabilities, which limits their practicality**.
> > > In summary, **entropy not only effectively reflects strategy execution difficulty but also provides a more accessible metric that requires no auxiliary models or interface support**, making it more practical and broadly applicable in real-world deployment scenarios.
> > >
> > >
> > > > **Q4.**   What temperature setting was used in the trials? Did you run trials with different temperatures?
> > >
> > > **A4.**  Thank you for the question. **All experiments in the main paper are conducted using the default temperature ($T = 1.0$)**. To further assess temperature sensitivity, we performed additional experiments on AIME 2025 with temperatures $T \in \{0.2, 0.6, 1.0, 1.4, 1.8\}$. The results show that performance varies moderately across temperatures: lower temperatures ($0.2$-$0.6$) yield more concentrated and deterministic reasoning paths, leading to reduced strategy diversity; higher temperatures ($1.4$-$1.8$) expand exploration within the model’s existing strategy space and produce more diverse reasoning paths, resulting in accuracy improvements from 16.6% to 23.3%.
> > >
> > > | Threshold | P@16 | P@32 | A@16 | A@32 |
> > > |-----------|------|------|------|------|
> > > | 0.2 | 33.3% | 36.6% | 20.0% | 20.0% |
> > > | 0.6 | 30.0% | 36.6% | 16.6% | 16.6% |
> > > | 1 | 26.6% | 33.3% | 13.3% | 16.6% |
> > > | 1.4 | 36.6% | 36.6% | 20.0% | 23.3% |
> > > | 1.8 | 36.6% | 36.6% | 23.3% | 23.3% |

---

> > > > ### Author Response · Authors · 2025-11-21
> > > >
> > > > > **Q5.**   Re. Appendix “A.4 EXPERIMENTAL VALIDATION OF THEOREM 3.5”: can you please share more details on the experimental setup involved (unless I missed it?)
> > > >
> > > > **A5.**  We appreciate the reminder. We have supplemented the details in Appendix A.4, and the revised version is as follows: To validate Theorem 3.5, we conduct experiments on a subset of the AQuA dataset using GPT-4o-mini. We first extract the set of recognizable strategies using the coarse-grained method (Uniform-C-Z), obtaining $m$ strategies in total. For each strategy, we sample 20 reasoning paths and compute strategy entropy based on the corresponding answer distributions. We denote the strategy with the lowest entropy as the lowest-complexity strategy $L$ and the one with the highest entropy as the highest-complexity strategy $H$.
> > > >
> > > > We then construct three sampling scenarios: (1) low-complexity–dominated sampling, (2) uniform sampling, and (3) high-complexity–dominated sampling. In each scenario, we generate 400 independent reasoning paths. In the low-complexity–dominated setting, $250/400$ of the sampling budget is allocated to strategy $L$, with the remaining budget distributed evenly across the other $(m-1)$ strategies. In the high-complexity–dominated setting, $250/400$ of the budget is allocated to strategy $H. In the uniform sampling setting, the budget is evenly allocated across all strategies. Performance is measured as the proportion of correct paths among the 400 samples.
> > > >
> > > > As shown in Appendix Figure 7, all samples exhibit a consistent trend: performance is highest when low-complexity strategies dominate, followed by uniform sampling, and lowest when high-complexity strategies dominate. This result aligns with the theoretical prediction of Theorem 3.5—that when the model favors low-complexity strategies, diversified sampling improves performance, whereas if the model favors high-complexity strategies, biased sampling significantly degrades performance.

---

> > > > > ### Comment · Reviewer_VoLR · 2025-11-27
> > > > > **ACK**
> > > > >
> > > > > Acknowledging other responses. Thanks for the additional explanation.

---

> > ### Comment · Reviewer_VoLR · 2025-11-27
> >
> > Regarding:
> > > ... generation parameters such as temperature only influence the degree of exploration within the existing strategy space and do not fundamentally alter the upper bound of the model’s strategy space...
> >
> > Do you have theoretical/empirical evidence pointing to this fact? I understand temperature directly impacts exploration within the strategy, but I believe you can do **fewer strategy samplings** if the model is allowed to be more creative.

---

> > > ### Author Response · Authors · 2025-11-28
> > >
> > > Thank you very much for your positive support of our paper!
> > >
> > > To address your question, we conducted validation experiments based on Uniform-F at AIME 2025. Specifically, we fixed (c''=3), set (c'=2/4), and kept the number of sampling steps constant at ($t$=12), while gradually increasing the temperature of GPT-4o-mini from 0.2 to 1.8. We then observed the number of identified strategies $m$, with the results summarized in the table below.
> > >
> > > | Temperature | 0.2 | 0.4 | 0.6 | 0.8 | 1 | 1.2 | 1.4 | 1.8 |
> > > |---------------|-----|-----|-----|-----|---|-----|-----|-----|
> > > | c'=2 (c''=3) | 4   | 5   | 6   | 6   | 8 | 9   | 8   | 8   |
> > > | c'=4 (c''=3) | 2   | 2   | 2   | 4   | 4 | 5   | 5   | 4   |
> > >
> > > The experimental results show that as the temperature rises, the model indeed identifies more strategies in the early stages. However, beyond approximately ($T$=0.8–1.2), the number of strategies no longer increases but instead stabilizes near an upper limit. This indicates that the effect of temperature primarily enhances the exploration of the existing internal strategy space. Once this space is sufficiently covered, further increasing the temperature hardly introduces new types of strategies. At the same time, **we also agree with your point: during the phase when exploration has not yet saturated, increasing the temperature can indeed reduce the budget required for strategy sampling**.
> > >
> > > Once again, we sincerely appreciate your constructive communication and positive feedback, and please feel free to reach out if you have any further questions.

---

### Official Review · Reviewer_Ercw · 2025-10-30

**Soundness:** 1
**Presentation:** 2
**Contribution:** 1
**Rating:** 2
**Confidence:** 4

**Summary:**

This paper shows that chain-of-thought (CoT) reasoning for large language models under test-time scaling (TTS) can suffer from strategy-selection bias, which limits multi-path exploration and reduces diversity. Theoretical analysis indicates that favoring low-complexity policies can be beneficial, whereas favoring high-complexity policies degrades TTS performance. To address this, the authors propose **TTS-Uniform**: (i) extract a set of candidate strategies; (ii) allocate the sampling budget uniformly across them; (iii) filter high-complexity policies using the entropy of the answer distribution; and (iv) aggregate the final predictions via majority voting. Experiments with GPT-family models on reasoning datasets show consistent improvements over baseline methods.

**Strengths:**

1. The paper highlights that strategy bias in the selection of reasoning paths during multi-path sampling is a key factor constraining path diversity.

2. It presents a fairly comprehensive theoretical analysis that characterizes how low- and high-complexity strategies affect model performance under test-time scaling (TTS).

**Weaknesses:**

1. **The approach introduces substantial additional overhead.** Both the coarse-grained and fine-grained variants depend on external (often stronger) models to extract strategies for every query (Q), which is impractical for real-world deployment.
2. **There is a gap between theory and method.** The theoretical analysis relies on the “Minimum Token Requirement of the Strategy” to derive conclusions about complexity and TTS performance, whereas the implementation uses entropy as a surrogate (owing to redundancy in CoT traces). Entropy is not within the prior theoretical framework, which considerably weakens the method’s theoretical underpinnings.
3. The entire empirical study is conducted on two closed-source GPT-series models, without using commonly adopted TTS research backbones such as the Qwen family. This raises concerns about the confidence and credibility of the experimental conclusions.
4. The experimental coverage is limited to a small set of mathematical reasoning datasets; key benchmarks like MATH are missing, and no general-purpose reasoning datasets are included, making it difficult to assess the method’s generalization.

**Questions:**

1. Multipath sampling faces a fundamental trade off between diversity and accuracy. For a given query Q with several plausible strategies, does distributing samples across them dilute the confidence assigned to the correct reasoning path? More extensive hyperparameter ablation studies are suggested to strengthen the empirical credibility of the results.
2. If concept level strategies are extracted solely through prompting, how can we ensure redundancy or strong correlation among the strategies? Does this practice shift policy bias from the target model to the external extractor model, with the latter’s bias merely being more favorable for solving the task rather than truly mitigating bias?

---

> ### Author Response · Authors · 2025-11-20
>
> We sincerely thank the reviewer for the valuable feedback and encouraging comments. We are motivated by the suggestions and have addressed each concern in detail as follows:
>
>
>
> > **W1.**   The approach introduces substantial additional overhead. Both the coarse-grained and fine-grained variants depend on external (often stronger) models to extract strategies for every query (Q), which is impractical for real-world deployment.
>
>
> **R1.** We thank the reviewer for the question regarding implementation details, **but this concern is based on a misunderstanding:** our strategy extraction process does not rely on external or stronger models; it is performed directly using the same model employed at inference time. The stronger models mentioned in Section 2 are used only to verify the existence of strategy bias in the evaluated model, rather than serving as components required during deployment.
>
> **(1) Overhead and budget constraints.**
> In all experiments and deployment settings, we strictly follow the constraints of test-time scaling: the total sampling budget is fixed (b), and we merely reallocate samples across strategies under this fixed budget. The number of final samples used for voting and decision-making never exceeds this budget. We acknowledge that organizing the strategy set requires several additional calls to the same model, but these calls only perform structural extraction rather than inference, incur lightweight overhead, and can be fully automated.
>
> **(2) Method objective and capability assumptions.**
> Our goal is to make explicit the set of strategies that the model has already internalized and can execute within its own training distribution, and then allocate sampling uniformly within this set to mitigate internal strategy bias. In other words, we address bias within the model’s own executable strategy space, rather than relying on a stronger model to generate new strategies beyond its knowledge boundary. Therefore, our method does not introduce external strategies, nor does it rely on a stronger model to improve reasoning ability.
>
>
> > **W2.**   There is a gap between theory and method. The theoretical analysis relies on the “Minimum Token Requirement of the Strategy” to derive conclusions about complexity and TTS performance, whereas the implementation uses entropy as a surrogate (owing to redundancy in CoT traces). Entropy is not within the prior theoretical framework, which considerably weakens the method’s theoretical underpinnings.
>
>
> **R2.** Thank you for pointing this out. We acknowledge the need to more explicitly connect the theoretical indicator $\ell_i$ with the observed entropy metric. Our logical chain is as follows: **a higher minimum token requirement $\ell_i$ implies a higher probability of reasoning errors due to cumulative deviation, which in turn leads to greater output uncertainty and thus higher entropy**. Accordingly, we do not treat entropy as a new indicator independent of the theory; rather, we view it as an observable proxy for the latent variable $\ell_i$. Meanwhile, the empirical results in Appendix A.5 (Figure 8) provide strong evidence supporting the validity of this proxy relationship.

---

> > ### Author Response · Authors · 2025-11-20
> >
> > > **W3 and W4.** The entire empirical study is conducted on two closed-source GPT-series models, without using commonly adopted TTS research backbones such as the Qwen family. This raises concerns about the confidence and credibility of the experimental conclusions. The experimental coverage is limited to a small set of mathematical reasoning datasets; key benchmarks like MATH are missing, and no general-purpose reasoning datasets are included, making it difficult to assess the method’s generalization.
> >
> > **R3.** We appreciate the reviewer’s valuable suggestion. Following your advice, we have added two additional open-source model families to complement the original GPT family: Qwen3-8B and LLaMA3-8B. Moreover, we have expanded the scope of evaluation tasks to further validate the cross-task generalization capability of our method. The newly included tasks consist of three mathematical reasoning benchmarks (HMMT_Feb_2025, AMC12_2024, MATH), one code reasoning task (MBPP), one symbolic reasoning task (ProntoQA), and one commonsense reasoning task (StrategyQA). For MATH, MBPP, ProntoQA, and StrategyQA, we evaluate on 100-sample subsets of each dataset. The total sampling budgets for all tasks are set to 16 and 32.
> >
> > | Dataset | Method | Qwen3-8b pass@16 | Qwen3-8b pass@32 | Qwen3-8b cons@16 | Qwen3-8b cons@32 | LLaMA3-8b pass@16 | LLaMA3-8b pass@32 | LLaMA3-8b cons@16 | LLaMA3-8b cons@32 |
> > |---------|--------|------------------|------------------|------------------|------------------|-------------------|-------------------|-------------------|-------------------|
> > | HMMT_feb_2025 | DeepConf | 3.3% | 3.3% | 0.0% | 0.0% | 6.7% | 10.0% | 0.0% | 3.3% |
> > | HMMT_feb_2025 | Our | 6.7% | 10.0% | 6.7% | 10.0% | 6.7% | 10.0% | 0.0% | 0.0% |
> > | AMC12_2024 | DeepConf | 0.0% | 6.7% | 0.0% | 2.2% | 20.0% | 31.1% | 8.8% | 11.1% |
> > | AMC12_2024 | Our | 11.1% | 11.1% | 4.4% | 4.4% | 28.9% | 33.3% | 26.3% | 28.9% |
> > | MATH | DeepConf | 31.0% | 42.0% | 14.0% | 16.0% | 56.0% | 68.0% | 26.0% | 29.0% |
> > | MATH | Our | 45.0% | 54.0% | 20.0% | 21.0% | 52.1% | 62.0% | 23.0% | 27.0% |
> > | MBPP | DeepConf | 35.0% | 35.0% | 30.0% | 30.0% | 28.0% | 28.0% | 22.0% | 28.0% |
> > | MBPP | Our | 43.0% | 50.0% | 26.0% | 30.0% | 54.0% | 54.0% | 24.0% | 28.0% |
> > | ProntoQA | DeepConf | 95.0% | 98.0% | 51.0% | 61.0% | 71.0% | 88.0% | 23.0% | 25.0% |
> > | ProntoQA | Our | 100.0% | 100.0% | 67.0% | 70.0% | 100.0% | 100.0% | 51.0% | 51.0% |
> > | StrategyQA | DeepConf | 76.0% | 94.0% | 48.0% | 49.0% | 57.0% | 71.0% | 22.0% | 28.0% |
> > | StrategyQA | Our | 87.0% | 96.0% | 57.0% | 63.0% | 83.0% | 83.0% | 30.0% | 30.0% |
> > | Avg. | DeepConf | 40.1% | 46.5% | 23.8% | 26.4% | 39.8% | 49.4% | 17.0% | 20.7% |
> > | Avg. | Our | 48.8% | 53.5% | 30.2% | 33.1% | 54.1% | 57.1% | 25.7% | 27.5% |
> >
> >
> >
> > We observe that our method consistently outperforms the latest TTS approach, DeepConf, across most tasks and model configurations. For instance, on Qwen3-8B, the average pass@16 increases from 40.1% to 48.8%, and pass@32 rises from 46.5% to 53.5%. This trend is particularly pronounced on mathematical reasoning tasks (e.g., HMMT_Feb_2025) and symbolic reasoning tasks (ProntoQA), indicating that the performance gains are not limited to specific models or task types. Instead, our approach improves reasoning consistency and final accuracy in a stable manner across both models and tasks.

---

> ### Author Response · Authors · 2025-11-20
>
> > **Q1.**   Multipath sampling faces a fundamental trade off between diversity and accuracy. For a given query Q with several plausible strategies, does distributing samples across them dilute the confidence assigned to the correct reasoning path? More extensive hyperparameter ablation studies are suggested to strengthen the empirical credibility of the results.
>
>
> **A1.**  In our setting, given a question and the set of strategies recognizable by the LLM, all of these strategies are in principle capable of leading to the correct answer. The differences among them do not lie in whether they are simply correct or incorrect, but rather in the varying stability of their reasoning processes induced by different levels of strategy complexity. Higher-complexity strategies typically contain more intermediate steps, making them less stable and more susceptible to deviation across repeated generations, ultimately resulting in higher error rates. According to our theoretical analysis, when the model implicitly favors low-complexity strategies, concentrating sampling on these strategies performs better than uniform sampling; however, when the model favors high-complexity strategies, concentrated sampling leads to a significant performance drop.
>
> Based on this, the effect of multi-path sampling on accuracy is not determined by the number of strategies per se, but by the complexity composition underlying the diversity. Specifically: **(1) when sampling is distributed across multiple low-complexity strategies, the overall error rate decreases and diversity improves performance; (2) when diversity mainly arises from high-complexity strategies, these paths tend to introduce additional noise, thereby degrading final performance.** We observe the same trend in our experiments on GPT-4o-mini and AQuA (see the table below).
>
> | Strategy | Sample 1 | Sample 2 | Sample 3 | Sample 4 | Sample 5 | Sample 6 |
> |----------|----------|----------|----------|----------|----------|----------|
> | Low      | 58.8%    | 58.5%    | 96.3%    | 23.0%    | 99.5%    | 51.8%    |
> | Uniform  | 55.5%    | 49.0%    | 92.5%    | 18.0%    | 99.0%    | 47.5%    |
> | High     | 52.7%    | 43.5%    | 89.3%    | 13.0%    | 98.5%    | 41.8%    |
>
> Therefore, **the key factor influencing performance is not whether diversity is increased, but which types of strategies the additional paths correspond to—that is, the complexity distribution underlying the diversity**.
>
>
>
> > **Q2.**   If concept level strategies are extracted solely through prompting, how can we ensure redundancy or strong correlation among the strategies? Does this practice shift policy bias from the target model to the external extractor model, with the latter’s bias merely being more favorable for solving the task rather than truly mitigating bias?
>
>
> **A2.**  We appreciate the reviewer’s insightful suggestion. Previously, we primarily relied on prompt examples and linguistic constraints to guide the model toward generating diverse strategies; however, this approach does not guarantee semantic-level deduplication. Therefore, we further introduce an explicit strategy deduplication step. After obtaining candidate strategies, we first generate a full reasoning path for each strategy and compute pairwise semantic similarity based on path representations (using MiniLM sentence embeddings with cosine similarity). We then iteratively remove one strategy from the most similar pair (chosen at random), and repeat this process until reaching the desired number of strategies (two strategies are removed in our experiments). On top of the deduplicated strategy set, we perform uniform sampling and entropy filtering.
>
> Empirically, this step improves the stability of reasoning outcomes. For example, on the AIME25 dataset under the same setting (A@32, three runs), the performance standard deviation decreases from 5.10 to 1.96.
>
> Finally, we emphasize that both the strategy extraction and the enhanced deduplication procedure are fully conducted using the same model without invoking any external models. Thus, **there is no phenomenon of transferring bias from the target model to any external model**. Further discussion on this point can be found in our response to W2.

---

> > ### Author Response · Authors · 2025-11-26
> > **Checking Whether Further Clarification Is Needed**
> >
> > Dear **Reviewer Ercw**,
> >
> > Thank you again for your detailed and constructive comments. We would like to kindly follow up to check whether our rebuttal and the additional analyses have fully addressed your concerns—specifically the clarification regarding implementation overhead and the misconception about using external models **(W1)**, the connection between the theoretical complexity indicator and the entropy surrogate **(W2)**, the expanded model families and broader empirical coverage including MATH and multiple general-purpose reasoning benchmarks **(W3–W4)**, as well as your questions on the diversity–accuracy trade-off and strategy redundancy **(Q1–Q2)**.
> >
> > If anything remains unclear or would benefit from further clarification, we would be very happy to provide additional details. Thank you once again for your time and thoughtful comments.
> >
> > Authors of **Submission 11524**

---

### Official Review · Reviewer_LbLj · 2025-10-30

**Soundness:** 2
**Presentation:** 3
**Contribution:** 2
**Rating:** 2
**Confidence:** 4

**Summary:**

Sampling-and-aggregation–based test-time scaling (TTS) methods suffer from substantial strategy-selection bias, which hampers efficient exploration of the solution space. This paper analyzes the impact of this bias on TTS effectiveness and introduces TTS-Uniform, a framework that (i) explicitly extracts candidate strategies, (ii) allocates the sampling budget uniformly across strategies, and (iii) filters potentially unstable or overly complex strategies via answer-entropy. Experiments on AQuA and AIME-2024/2025, using two OpenAI small models, show consistent improvements over standard TTS baselines.

**Strengths:**

1. **Clear and intuitive problem formulation.** The paper formalizes “strategies” via equivalence classes and empirically demonstrates strategy-selection bias through clustering and visualizations, offering an intuitive lens for analyzing TTS.
2. **Simple yet effective approach.** Uniform budget allocation combined with answer-entropy–based pruning is straightforward to implement and yields more pronounced improvements on weaker models.
3. **Insightful empirical analysis.** The finding that weaker models tend to exhibit stronger bias and consequently benefit more which is coherently explained using distribution-level comparisons of expected error rates; the argument is simple but illuminating.

**Weaknesses:**

1. **Strong yet insufficiently validated core assumption.** The method treats the *minimum token requirement* as a complexity measure and assumes a monotone positive correlation with error rate; however, the evidence provided is largely intuitive.

2. **Overly general theoretical result.** Under monotonicity, *Theorem 3.5* effectively reduces to a first-order stochastic dominance (FOSD) statement. The analysis does not formally connect majority-vote aggregation or entropy-based pruning to the resulting Acc@k. Recommended:

   * Derive lower/upper bounds for majority-vote accuracy under with/without bias and dependent/independent error assumptions when using uniform sampling + entropy pruning;
   * Characterize how intra-strategy versus inter-strategy error correlation modulates voting gains.

3. **Questionable use of entropy as a complexity proxy.** High answer entropy may stem from task ambiguity or a larger output space rather than intrinsically “complex” strategies. Please add (i) statistical experiments, (ii) comparisons against other proxy choices (e.g., minimal provable path-length estimates, step consistency), and (iii) an expanded discussion of the hyperparameter n.

4. **Methodological detail gaps.** (i) Uniform-F requires explicit specification of the node-similarity metric, merge threshold, and denoising rules, as well as their effects on the number of strategies (m) and coverage; (ii) provide a sensitivity/cost analysis for hyperparameters—total budget (b), pre-sampling (t, i.e., strategy-set size), and the number of discarded strategies (n).

5. **Limited experimental breadth and diagnostics.** (i) Baselines: include recent representative TTS methods (e.g., *DeepConf*, etc.) to strengthen the claims; (ii) Benchmarks: extend beyond a small set of math reasoning tasks to HMMT/AMC and code/symbolic reasoning to assess generality; (iii) Diagnosing strategy-selection bias: beyond the cluster distribution in Fig. 2, report relationships to complexity and accuracy, and run rigorous ablations to isolate true gains from strategy diversity.

**Questions:**

1. To what extent does the assumed complexity–error monotonicity generalize across models and tasks? Please provide quantitative evidence across model families and multiple benchmarks.
2. How is the entropy-pruning threshold determined?
3. For Uniform-F, what are the explicit rules for selection, expansion, simulation, and backpropagation?
4. How robust are the bias estimates to the choice of evaluator model and the target model under evaluation? Do the conclusions change under alternative evaluators or targets?

---

> ### Author Response · Authors · 2025-11-20
>
> We sincerely thank the reviewer for the valuable feedback and encouraging comments. We are motivated by the suggestions and have addressed each concern in detail as follows:
>
> > **W1.**   **Strong yet insufficiently validated core assumption.** The method treats the minimum token requirement as a complexity measure and assumes a monotone positive correlation with error rate; however, the evidence provided is largely intuitive.
>
> **R1.**  Thank you for pointing this out. We base this hypothesis on the probabilistic nature of LLM autoregressive generation. Suppose strategy $\mathcal{R}i$ requires logical steps $s_1,\dots,s_k$, where $k$ is proportional to the minimum token requirement $\ell_i$. Since each step generation has a non-zero error probability $\varepsilon_i$, assuming the error probabilities at each step are independent, then the overall correctness probability is $\Pr[\text{correct}] \approx \prod{i\in [k]}(1-\varepsilon_i)^k\approx (1-\bar{\epsilon})^k$, where $\overline{\varepsilon}$ is some average of $\varepsilon_1,\cdots,\varepsilon_k$, which means the overall error rate $1-(1-\bar{\epsilon})^k$ increases monotonically with $k$, i.e., with $\ell_i$.
>
> Due to the complex correlations between reasoning steps, but reasoning steps are often positively correlated, meaning subsequent steps depend on previous ones, this correlation leads to a higher probability of error in the next step given an error in the current step, thus causing errors to accumulate, and we have $\Pr[\mathrm{correct}] > \prod_{i\in [k]}(1-\varepsilon_i)$; in addition, the model exhibits self-correction behavior, but in complex and rigorous reasoning, the probability of self-correction is usually weaker than the error probability, and the process of correction leads to an increase in the number of tokens in the response, generating many redundant logical chains. Therefore, the complex correlations actually strengthen rather than weaken the independence assumption.

---

> > ### Author Response · Authors · 2025-11-20
> >
> > > **W2.**   **Overly general theoretical result.** Under monotonicity, Theorem 3.5 effectively reduces to a first-order stochastic dominance (FOSD) statement. The analysis does not formally connect majority-vote aggregation or entropy-based pruning to the resulting Acc@k. Recommended: **a)** Derive lower/upper bounds for majority-vote accuracy under with/without bias and dependent/independent error assumptions when using uniform sampling + entropy pruning; **b)** Characterize how intra-strategy versus inter-strategy error correlation modulates voting gains.
> >
> > **R2.**  We thank you for this valuable and constructive feedback.
> >
> > **a. Accuracy Upper and Lower Bounds:**
> >
> > We first consider the simplified scenario with intra-strategy error consistency and inter-strategy independence, which is essentially captured by the following assumptions:
> >
> > - **Assumption A (Intra-strategy)**: The errors of the $b/m$ paths sampled from the same strategy $R_i$ are **perfectly correlated**. This means that the $b/m$ votes from the same strategy can be viewed as a single voter $v_i$, whose accuracy is the strategy's accuracy $(1 - \epsilon_i)$.
> >
> > - **Assumption B (Inter-strategy)**: The errors between different policies (i.e., between voters $v_i$ and $v_j$) are **perfectly independent**.
> >
> > Under these assumptions, the model (uniform sampling + majority voting) simplifies to $m$ independent voters $v_1, \dots, v_m$ with accuracies $p_1, \dots, p_m$, respectively. Suppose we sort the voters in descending order of error rate, i.e., $\epsilon_1 \ge \epsilon_2 \ge \dots \ge \epsilon_m$ (where $v_1$ is the worst and $v_m$ is the best).
> >
> > The effect of entropy pruning is to discard $n$ voters and retain $K = m-n$ voters for majority voting. Let $S$ be the set of retained voters. The final accuracy is the probability that the majority voting succeeds, i.e., $\Pr\left[\frac{1}{K}\sum_{j \in S} v_j > 0.5\right]$.
> >
> > - **Upper Bound**: The best case is to discard the $n$ strategies with the highest error rates (i.e., $v_1, \dots, v_n$), and retain the best-performing set $S_{\text{best}} = \{v_{n+1}, \dots, v_m\}$. The accuracy upper bound is $\Pr\left[\frac{1}{K}\sum_{j \in S_{\mathrm{best}}} v_j > 0.5\right]$.
> > - **Lower Bound**: The worst case is to discard the $n$ strategies with the lowest error rates (i.e., $v_{m-n+1}, \dots, v_m$), and retain the worst-performing set $S_{\text{worst}} = \{v_1, \dots, v_{m-n}\}$. The accuracy lower bound is $\Pr\left[\frac{1}{K}\sum_{j \in S_{\text{worst}}} v_j > 0.5\right]$.
> >
> > Since we have empirically demonstrated in Appendix A.5 that entropy is a valid proxy for accuracy, our "entropy pruning" strategy effectively identifies low-quality policies in practice, approaching the theoretical upper bound.
> >
> >
> >
> > **b. Impact of Error Correlation:**
> >
> > We first assume that an oracle provides the intra-strategy correlation coefficient $\rho_{\text{intra}}$ and the inter-strategy correlation coefficient $\rho_{\text{inter}}$ (with $\rho_{\text{intra}} > \rho_{\text{inter}}$).
> >
> > I believe the question here is: Consider a set of $M$ groups of voters, each group containing $N$ voters. Each voter is a random variable $v_{ij} \sim \mathrm{Ber}(p)$. The correlation between $v_{ij}$ and $v_{ij'}$ is $\rho_{\mathrm{in}}$, and the correlation between $v_{ij}$ and $v_{i'j'}$ (for $i \neq i'$) is $\rho_{\mathrm{out}}$ (with $\rho_{\mathrm{out}} < \rho_{\mathrm{in}}$). Let $k = MN$, and define $r = \frac{1}{k}\sum_{i=1}^{M}\sum_{j=1}^{N} v_{ij}$. Then, $\mathrm{Acc}@k = \Pr[r > 1/2]$. We aim to compute the relationship between $\mathrm{Acc}@k$ and $\rho_{\mathrm{in}}$, $\rho_{\mathrm{out}}$.
> >
> > By the central limit theorem, $r$ can be approximated as following a normal distribution $\mathcal{N}(\mathbb{E}[r], \mathsf{Var}[r])$. Here, $\mathbb{E}[r] = \frac{1}{k}\sum_{i=1}^{M}\sum_{j=1}^{N} \mathbb{E}[v_{ij}] = p$. Let $\sigma^2 = \mathsf{Var}[v_{ij}] = p(1-p)$. Then, the variance of $r$ is:
> > \begin{equation}
> > \mathrm{Var}[r] = \frac{1}{k^2} \left( k\sigma^2 + k(N-1)\rho_{\mathrm{in}}\sigma^2 + k(k-N)\rho_{\mathrm{out}}\sigma^2 \right) = \frac{\sigma^2}{k} \left( 1 + (N-1)\rho_{\mathrm{in}} + (k-N)\rho_{\mathrm{out}} \right)
> > \end{equation}
> >
> > Therefore,
> > \begin{equation}
> > \mathrm{Acc}@k = \Pr[r > 0.5] \approx \Pr_{g \sim \mathcal{N}(0,1)} \left[ g > \frac{0.5 - \mathbb{E}[r]}{\sqrt{\mathrm{Var}[r]}} \right] = \Phi \left( \frac{\sqrt{k} (p - 0.5)}{\sigma \sqrt{1 + (N-1)\rho_{\mathrm{in}} + (k-N)\rho_{\mathrm{out}}}} \right)
> > \end{equation}
> > From this, it can be seen that smaller correlations lead to larger $\mathrm{Acc}@k$. Moreover, when $M \ge 2$, we have $k - N > N - 1$, so the inter-group correlation has a greater impact on $\mathrm{Acc}@k$ than the intra-group correlation.
> >
> > However, modeling the error correlation specifically for LLMs is an open problem that depends on the LLM architecture and task complexity, and we leave it as future work.

---

> > > ### Author Response · Authors · 2025-11-20
> > >
> > > > **W3.**   Questionable use of entropy as a complexity proxy. High answer entropy may stem from task ambiguity or a larger output space rather than intrinsically “complex” strategies. Please add (i) statistical experiments, (ii) comparisons against other proxy choices (e.g., minimal provable path-length estimates, step consistency), and (iii) an expanded discussion of the hyperparameter n.
> > >
> > > **R3.**   Thank you for raising this insightful question. We agree that task ambiguity and output space size affect absolute entropy. However, given a fixed question, these factors remain constant across strategies and thus do not alter relative differences. Under this setting, entropy primarily reflects execution stability: strategies requiring more intermediate conditions or exhibiting weaker robustness lead to higher variance across repeated generations (thus higher entropy), while simpler, more stable strategies yield concentrated outputs. Therefore, using entropy as a proxy for strategy complexity (execution stability) is reasonable.
> > >
> > >
> > > ### **(i & ii) Statistical validation and comparison with alternative proxies**
> > >
> > > To further validate the use of entropy as a proxy for strategy complexity, we conduct a comparative analysis against alternative complexity measures (i.e., estimated minimum path length and step consistency). Specifically, using GPT-4o-mini, we construct experimental samples on AIME 2025 and AQuA with a total sampling budget of 32 per question. For each question, we extract 4 strategies and generate 8 samples per strategy.
> > >
> > > For the estimated minimum path length, we employ a stronger reasoning model (OpenAI-o3) to summarize and abstract the reasoning chains under each strategy, yielding an estimated minimal path length for that strategy. For step consistency, we encode all reasoning paths from the same strategy using a sentence-level encoder (MiniLM in our experiment), compute pairwise cosine similarity between embeddings, and take the average similarity as the consistency score.
> > >
> > > Finally, we measure strategy accuracy as the proportion of correct reasoning paths among all sampled paths for that strategy. The quantitative results and comparative analysis are reported in the table below.
> > >
> > > | AIME 2025 | Strategy 1 | Strategy 2 | Strategy 3 | Strategy 4 |
> > > |-----------|------------|------------|------------|------------|
> > > | Answer Entropy | 2.08 | 1.91 | 1.67 | 2.08 |
> > > | Step Consistency | 0.8 | 0.92 | 0.87 | 0.78 |
> > > | Min Tokens | 80 | 73 | 64 | 103 |
> > > | Accuracy | 0% | 25.0% | 37.5% | 0% |
> > >
> > > | AQuA | Strategy 1 | Strategy 2 | Strategy 3 | Strategy 4 |
> > > |------|------------|------------|------------|------------|
> > > | Answer Entropy | 1.07 | 0.38 | 0.9 | 1.49 |
> > > | Step Consistency | 0.92 | 0.96 | 0.88 | 0.89 |
> > > | Min Tokens | 68 | 52 | 83 | 80 |
> > > | Accuracy | 62.5% | 87.5% | 62.5% | 37.5% |
> > >
> > > Subsequently, we compute the Pearson correlation coefficients between the three strategy complexity metrics (answer entropy, estimated minimum path length, and step consistency) and strategy accuracy, in order to quantify the strength of association between each complexity measure and performance. The results are summarized in the table below.
> > >
> > > | Dataset | Entropy | Consistency | Min Tokens |
> > > |---------------------|---------|-------------|------------|
> > > | AIME 25             | -0.968  | 0.819       | -0.826     |
> > > | AQuA                | -0.986  | 0.795       | -0.812     |
> > > | Avg                 | -0.977  | 0.807       | -0.819     |
> > >
> > >
> > > Entropy shows the strongest and most consistent correlation with accuracy (|r| = 0.977), higher than consistency and token length. Minimum token count is less reliable due to summarization noise; step consistency is effective but requires extra encoding and pairwise similarity, increasing overhead. Entropy requires no auxiliary model, making it both **effective and practical**.
> > >
> > > ### **(iii) Sensitivity of hyperparameter n**
> > >
> > > We vary pruning ratios by removing the top-entropy strategies (k ∈ {12.5–87.5%}, n = 1–7) under a fixed budget of 32.
> > >
> > > | Dataset  | n=1   | n=2   | n=3       | n=4       | n=5       | n=6   | n=7   |
> > > | -------- | ----- | ----- | --------- | --------- | --------- | ----- | ----- |
> > > | AQuA     | 89.0% | 89.4% | **89.8%** | 89.0%     | 87.8%     | 87.4% | 87.4% |
> > > | AIME 2025 | 3.3%  | 6.6%  | 13.3%     | **16.6%** | **16.6%** | 13.3% | 10.0% |
> > >
> > >
> > > The results show that model performance remains largely stable across different pruning ratios k (corresponding to the number of high-complexity strategies removed n), with the optimal range observed at n = 3 to 5: AQuA achieves 89.8% at n = 3, while AIME 2025 reaches 16.6% at n = 4 and 5. Insufficient pruning (n = 1) retains too many unstable strategies, while excessive pruning (n = 7) may remove stable strategies incorrectly, both leading to performance degradation. Therefore, the method is robust and insensitive to the hyperparameter, and adjusting k around 50% generally yields near-optimal results.

---

> > > ### Comment · Reviewer_LbLj · 2025-11-27
> > >
> > > I have two follow-ups:
> > >
> > > 1. Using entropy as the indicator essentially measures model confidence. Under the paper’s logic, entropy is adopted, the “minimum-token” assumption becomes redundant aside questions about its correctness or validity.
> > >
> > > 2. There is also a tension between confidence and strategy debiasing: if higher-confidence strategies tend to be more accurate, should we sample uniformly across strategies or allocate more budget to high-confidence strategies? Of course, retaining higher-confidence paths within the same strategy is unproblematic.

---

> > ### Comment · Reviewer_LbLj · 2025-11-27
> >
> > Thank you for the detailed response.
> >
> > 1. The independence assumption across autoregressive steps does not hold in LLM reasoning. Error probabilities between steps are correlated, so an independence-based derivation is not justified.
> >
> > 2. The preconditions under which strategy-debiasing or the minimum-token hypothesis would yield benefits are rarely met in practice. Such benefits presuppose that, for a given question, the model exhibits multiple correct solution paths under distinct strategies that are observable under finite sampling. When the per-strategy correctness probabilities are approximately flat, debiasing can improve robustness. In practice, however, models typically solve an instance via a single viable strategy; other strategies have an effectively infinite minimum-token requirement. When the correctness distribution across strategies is sharply peaked, the right objective is to identify high-quality strategies to improve accuracy rather than to debias.
> >
> > 3. Consequently, strategy debiasing should be model-specific; strategy diversity is not a necessary condition for accurate reasoning. Moreover, the per-strategy accuracy distribution is often sharply peaked, so debiasing does not provide consistent gains.

---

> ### Author Response · Authors · 2025-11-20
>
> > **W4, Q2, and Q3.**   **Methodological detail gaps.** (i) Uniform-F requires explicit specification of the node-similarity metric, merge threshold, and denoising rules, as well as their effects on the number of strategies (m) and coverage; (ii) provide a sensitivity/cost analysis for hyperparameters—total budget (b), pre-sampling (t, i.e., strategy-set size), and the number of discarded strategies (n).
>
> **R4.**  (i) We thank the reviewer for the suggestion and have added more details about Uniform-F. We first generate t pre-sampled reasoning paths \(t < b\) for each problem and segment them into steps. Then, we use a large language model to summarize each step into abstract representations by condensing specific computational processes into higher-level operation types with one or two sentences, making nodes comparable across paths. Based on this, we employ **LLMs and prompt engineering** to align and match abstract nodes, treating nodes categorized into the same operational pattern as equivalent, and extract recurring contiguous sub-chains across paths.
>
> To ensure strategy stability, we introduce two thresholds during candidate sub-chain construction: **the minimum support threshold c'** requires candidate sub-chains to appear in at least c' different reasoning paths to avoid coincidental patterns; **the minimum node threshold c''** filters out sub-chains shorter than c''. **Larger c' and c'' values retain more stable and consistent mainstream strategies, resulting in fewer final strategies m but lower coverage**. Conversely, **smaller thresholds preserve more fine-grained strategies, increasing m and coverage while being more prone to noise**. The table below shows the number of strategies obtained with different c' and c'' combinations (GPT-4o-mini, AIME 2025).
>
> | c' \ c" | Two | Three | Four | Five |
> |---------|---|---|---|---|
> | Two       | 9 | 8 | 6 | 6 |
> | Three       | 7 | 7 | 5 | 4 |
> | Four       | 6 | 4 | 3 | 2 |
> | Five       | 4 | 3 | 2 | 2 |
>
>
> (ii) We further analyze the sensitivity of Uniform-F to three hyperparameters: the total budget b, the pre-sampling size t, and the number of pruned strategies n (expressed as the pruning ratio k). As shown in Table, b is the primary factor affecting performance. With fixed k=50% and t=12, increasing b from 32 to 64 improves the accuracy on AIME 2025 from 13.3% to 20.0%. In contrast, the influence of t is relatively weaker, likely because the current range of t already covers most strategic patterns, and further increases yield only marginal gains. The effect of k is moderate: pruning too little retains excessive unstable strategies, while pruning too much may incorrectly remove stable strategies, both leading to performance degradation. Therefore, **selecting k around 50% generally provides a robust and balanced outcome**.
>
> | Budget (b) | t=10 (k=50%) | t=12 (k=50%) | t=14 (k=50%) | k=25% (t=12) | k=50% (t=12) | k=75% (t=12) |
> |------------|---------------|---------------|---------------|---------------|---------------|---------------|
> | 32         | 13.3%         | 13.3%         | 10.0%         | 10.0%         | 13.3%         | 10.0%         |
> | 64         | 20.0%         | 20.0%         | 20.0%         | 16.6%         | 20.0%         | 20.0%         |

---

> > ### Author Response · Authors · 2025-11-20
> >
> > > **W5.**   **Limited experimental breadth and diagnostics.** (i) Baselines: include recent representative TTS methods (e.g., DeepConf [1], etc.) to strengthen the claims; (ii) Benchmarks: extend beyond a small set of math reasoning tasks to HMMT/AMC and code/symbolic reasoning to assess generality; (iii) Diagnosing strategy-selection bias: beyond the cluster distribution in Fig. 2, report relationships to complexity and accuracy, and run rigorous ablations to isolate true gains from strategy diversity.
> >
> >
> > **R5.**  Thank you for pointing this out.
> >
> > **(i)** and **(ii)**: Following your comments, we include DeepConf, a recent test-time scaling approach, as an additional baseline and expand our evaluation tasks to further assess cross-task generalization. The newly added benchmarks include two mathematical reasoning tasks (HMMT_Feb_2025 and AMC12_2024), one code reasoning task (MBPP), and one symbolic reasoning task (ProntoQA), where MBPP and ProntoQA are evaluated using their respective 100-sample subsets. We compare our method against DeepConf using two open-source models, Qwen3-8B and LLaMA3-8B, with total sampling budgets set to 16 and 32, respectively. The complete experimental results are summarized in the Table below.
> >
> > | Dataset | Method | Qwen3-8b pass@16 | Qwen3-8b pass@32 | Qwen3-8b cons@16 | Qwen3-8b cons@32 | LLaMA3-8b pass@16 | LLaMA3-8b pass@32 | LLaMA3-8b cons@16 | LLaMA3-8b cons@32 |
> > |---------|--------|------------------|------------------|------------------|------------------|-------------------|-------------------|-------------------|-------------------|
> > | HMMT_feb_2025 | DeepConf | 3.3% | 3.3% | 0.0% | 0.0% | 6.7% | 10.0% | 0.0% | 3.3% |
> > | HMMT_feb_2025 | Our | 6.7% | 10.0% | 6.7% | 10.0% | 6.7% | 10.0% | 0.0% | 0.0% |
> > | AMC12_2024 | DeepConf | 0.0% | 6.7% | 0.0% | 2.2% | 20.0% | 31.1% | 8.8% | 11.1% |
> > | AMC12_2024 | Our | 11.1% | 11.1% | 4.4% | 4.4% | 28.9% | 33.3% | 26.3% | 28.9% |
> > | MBPP | DeepConf | 35.0% | 35.0% | 30.0% | 30.0% | 28.0% | 28.0% | 22.0% | 28.0% |
> > | MBPP | Our | 43.0% | 50.0% | 26.0% | 30.0% | 54.0% | 54.0% | 24.0% | 28.0% |
> > | ProntoQA | DeepConf | 95.0% | 98.0% | 51.0% | 61.0% | 71.0% | 88.0% | 23.0% | 25.0% |
> > | ProntoQA | Our | 100.0% | 100.0% | 67.0% | 70.0% | 100.0% | 100.0% | 51.0% | 51.0% |
> > | Avg. | DeepConf | 33.3% | 35.7% | 20.3% | 23.3% | 31.4% | 39.3% | 13.5% | 16.9% |
> > | Avg. | Our | 40.2% | 42.8% | 26.0% | 28.6% | 47.4% | 49.3% | 25.3% | 27.0% |
> >
> >
> > We observe that our method significantly outperforms DeepConf across most tasks and model settings. For example, on Qwen3-8B, the average pass@16 improves from 33.3% to 40.2%, and pass@32 improves from 35.7% to 42.8%. This trend is particularly pronounced on mathematical reasoning tasks (e.g., HMMT_Feb_2025) and symbolic reasoning tasks (ProntoQA), indicating that the performance gains are not limited to a specific task type, but instead consistently enhance reasoning stability and final accuracy across diverse tasks.
> >
> >
> > **(iii)**: To determine whether the performance gain comes from strategy diversity or from filtering and distribution correction, we conduct three controlled experiments on AIME 2025 and AQuA using GPT-4o-mini. (1) Uniformly sample all extracted strategies and discard n high-entropy strategies, then vote using the remaining strategies. (2) Uniformly sample all strategies and randomly remove n strategies, keeping the same number as in (1). (3) Uniformly sample all strategies without filtering. As shown in the table, entropy-based filtering achieves the best performance, no-filter uniform sampling ranks second, and random removal performs worst. This indicates that the improvement is not due to increased diversity, but to filtering that amplifies low-complexity strategies while suppressing high-complexity ones, leading to more stable reasoning.
> >
> >
> > | Method       | AIME 2025 P@32 | AIME 2025 P@64 | AIME 2025 A@32 | AIME 2025 A@64 | AQuA P@32 | AQuA P@64 | AQuA A@32 | AQuA A@64 |
> > |--------------|----------------|----------------|----------------|----------------|-----------|-----------|-----------|-----------|
> > | m-n (our)    | 33.3%          | 40.0%          | 16.6%          | 20.0%          | 95.3%     | 97.6%     | 89.0%     | 89.8%     |
> > | m-n (random) | 23.3%          | 33.3%          | 13.3%          | 13.3%          | 91.7%     | 93.0%     | 88.2%     | 88.5%     |
> > | Uniform      | 33.3%          | 40.0%          | 16.6%          | 16.6%          | 96.1%     | 97.6%     | 89.0%     | 89.0%     |
> >
> > [1] Fu Y, Wang X, Tian Y, et al. Deep think with confidence[J]. arXiv preprint arXiv:2508.15260, 2025.

---

> > > ### Author Response · Authors · 2025-11-20
> > >
> > > > **Q1.**   To what extent does the assumed complexity–error monotonicity generalize across models and tasks? Please provide quantitative evidence across model families and multiple benchmarks.
> > >
> > > **A1.** We thank the reviewer for the helpful suggestion. To examine whether the relationship between complexity and error rate generalizes across models and tasks, we compute the Pearson correlation between strategy entropy and strategy accuracy on three model families (GPT-4o-mini, Qwen3-8B, LLaMA3-8B) and four task types (mathematics: AIME 2025; code: MBPP; symbolic reasoning: ProntoQA; commonsense reasoning: StrategyQA). The results show a strong and consistent negative correlation across models and tasks, with an average magnitude of approximately |r| ≈ 0.84, indicating that higher-complexity strategies are generally more error-prone. This demonstrates that the relationship holds not only within a single task, but generalizes across architectures and task types, supporting our choice of entropy as a proxy for strategy complexity.
> > >
> > > | Pearson Correlation | AIME 2025 | MBPP | ProntoQA | StrategyQA |
> > > |---------------------|-----------|------|----------|------------|
> > > | GPT-4o-mini         | -0.97     | -0.92 | -0.85    | -0.86      |
> > > | Qwen3-8b            | -0.73     | -0.89 | -0.84    | -0.69      |
> > > | LLaMA3-8b           | -0.85     | -0.79 | -0.93    | -0.79      |
> > > | Avg                 | -0.85     | -0.87 | -0.87    | -0.78      |
> > >
> > >
> > > > **Q4.**   How robust are the bias estimates to the choice of evaluator model and the target model under evaluation? Do the conclusions change under alternative evaluators or targets?
> > >
> > >
> > > **A4.**  In the table below, we keep the evaluator model fixed (OpenAI-o3) and replace the model being evaluated from GPT-4o-mini to OpenAI-o1-mini and Gemini-2.5-Flash. The results show substantial shifts in strategy distributions; for instance, strategy 1 drops from 52.4% (GPT-4o-mini) to 22.1% (Gemini-2.5-Flash), while strategy 3 increases from 19.0% to 26.7%. This indicates that different models exhibit systematic differences in strategy preferences due to varying training data and internal inductive biases—that is, strategy bias is an inherent property of a model, and different models may favor different strategies.
> > >
> > > | Model | Strategy 1 | Strategy 2 | Strategy 3 | Strategy 4 |
> > > |-------|------------|------------|------------|------------|
> > > | GPT-4o-mini | 52.4% | 26.8% | 19.0% | 1.8% |
> > > | OpenAI-o1-mini | 40.3% | 20.3% | 21.7% | 0.2% |
> > > | Gemini-2.5-Flash | 22.1% | 5.9% | 26.7% | 10.6% |
> > >
> > > In contrast, in the table below, we keep the model being evaluated fixed (GPT-4o-mini) and vary only the evaluator model (GPT-4o, Gemini-Flash). In this setting, the overall structure of the strategy distribution remains consistent with only minor fluctuations, indicating that the evaluator does not fundamentally alter the true strategy distribution of the evaluated model. The small differences arise because different evaluators possess different prior knowledge and may judge a few strategies differently; however, the overall ranking of strategy preferences remains consistent.
> > >
> > > | Evaluation Model | Strategy 1 | Strategy 2 | Strategy 3 | Strategy 4 |
> > > |------------------|------------|------------|------------|------------|
> > > | OpenAI-o3        | 52.4%      | 26.8%      | 19.0%      | 1.8%       |
> > > | GPT-4o           | 47.7%      | 30.9%      | 16.4%      | 0.7%       |
> > > | Gemini-2.5-Flash | 42.2%      | 18.1%      | 28.6%      | 2.0%       |

---

> > > > ### Author Response · Authors · 2025-11-26
> > > > **Checking Whether Further Clarification Is Needed**
> > > >
> > > > Dear **Reviewer LbLj**,
> > > >
> > > > Thank you again for your detailed and constructive feedback. We would like to kindly follow up to check whether our rebuttal and the additional analyses have fully addressed your concerns—specifically the validation of the assumption **(W1)**, the theoretical bounds and correlation characterization **(W2)**, the statistical comparisons and entropy-proxy justification **(W3)**, the methodological specifications and sensitivity analysis **(W4)**, and the expanded experiments and diagnostics **(W5)**, as well as your additional questions.
> > > >
> > > > If anything remains unclear or would benefit from further clarification, we would be very happy to provide additional details. Thank you once again for your time and thoughtful comments.
> > > >
> > > > Authors of **Submission 11524**

---

> > > > ### Comment · Reviewer_LbLj · 2025-11-27
> > > >
> > > > Thank you for the additional detailed experiments. I recommend strengthening the paper’s logical rigor and tightening the theoretical analysis. If you continue to frame the work around strategy debiasing, please delineate the applicability conditions and model regimes where debiasing is appropriate, and provide clear, well-justified theory and experiments.
> > > >
> > > > On methodology, please add concrete details: clarify whether strategy extraction is performed by the target model itself or by an auxiliary LLM; specify the Uniform-F algorithm (selection, expansion, merging rules) and its hyperparameters; and evaluate under cost-matched settings against strong baselines. Finally, choose appropriate backbone models and datasets to ensure the conclusions are robust and broadly applicable.

---

> > > > > ### Author Response · Authors · 2025-11-28
> > > > >
> > > > > **Thank you very much for taking the time to engage with our work through such constructive communication and for raising a series of highly insightful and thought-provoking questions. We sincerely appreciate your effort and enthusiasm. We address each of your points in detail below:**
> > > > >
> > > > > > **New_Q1.**   The independence assumption across autoregressive steps does not hold in LLM reasoning. Error probabilities between steps are correlated, so an independence-based derivation is not justified.
> > > > >
> > > > > **New_A1.** Thank you for your question. We will further provide conclusions for the case when reasoning steps are correlated (as we briefly explained in our previous response to weakness W1, **our hypothesis still holds even when there are complex correlations between different steps**).
> > > > >
> > > > > Specifically, we assume there are $k$ reasoning steps. Let event $E_i$ denote whether the $i$-th step is correct, taking values $0$ (incorrect) or $1$ (correct). We demonstrate that $\Pr\left[\bigwedge_{i\in[k]}E_i=1\right]$ decreases with $k$.
> > > > >
> > > > > We only consider valid proofs, i.e., when computing $\Pr[E_{i+1}=1|\text{some }j<i,E_j=0]$, we consider the probability of:
> > > > > (1) correcting all errors in the previous steps (that were not corrected in the first $i$ steps) at step $i+1$; and
> > > > > (2) the $i+1$-th step being correct.
> > > > >
> > > > > Let $A_n=\bigcap_{i\in [n]}E_i$, then $A_n\subseteq A_{n-1}$. Since $\Pr[S]\le \Pr[T]$ when $S\subseteq T$, we have $\Pr[A_n]\le \Pr[A_{n-1}]$. **Therefore, $\Pr[A_n]$ monotonically decreases with $n$. This derivation holds regardless of whether the events are positively correlated, negatively correlated, or independent**. This means that the more reasoning steps there are, the lower the correctness rate, which indicates a negative correlation between the minimum token requirement and the correctness rate.
> > > > >
> > > > >
> > > > > > **New_Q2.**   The preconditions under which strategy-debiasing or the minimum-token hypothesis would yield benefits are rarely met in practice. Such benefits presuppose that, for a given question, the model exhibits multiple correct solution paths under distinct strategies that are observable under finite sampling. When the per-strategy correctness probabilities are approximately flat, debiasing can improve robustness. In practice, however, models typically solve an instance via a single viable strategy; other strategies have an effectively infinite minimum-token requirement. When the correctness distribution across strategies is sharply peaked, the right objective is to identify high-quality strategies to improve accuracy rather than to debias.
> > > > >
> > > > > **New_A2.** We thank the reviewer for the insightful comments. Our experiments show that across diverse tasks—including mathematics, symbolic reasoning, and code reasoning—our method consistently improves performance across different model families. **This indicates that the feasible strategy space of LLMs is in fact richer than intuition might suggest, and thus mitigating the adverse effects of strategy bias at test time is broadly beneficial in practice**.
> > > > >
> > > > > More importantly, our method does not rely on the idealized assumption that “all strategies have roughly similar correctness rates.” Instead, it is built upon a weaker and more realistic premise: high-complexity strategies tend to be less stable and typically exhibit higher answer entropy. **The roles of uniform sampling and entropy filtering are therefore not merely to debias strategy selection**, but to suppress instability when the model favors high-complexity strategies, while preserving the beneficial impact of low-complexity strategies when they are preferred. **As a result, the method effectively identifies and reinforces high-quality (low-complexity) strategies, which is particularly crucial when the correctness distribution is sharply peaked**.
> > > > >
> > > > > It is also worth noting that the reviewer’s observation—that “the model often relies on a single feasible strategy, while other strategies have effectively infinite minimum-token requirements”—precisely corresponds to the scenario where the model strongly favors low-complexity strategies. In such cases, secondary strategies tend to be less stable and produce answers with higher entropy, and thus are naturally filtered out by our entropy-based mechanism; meanwhile, the dominant low-complexity strategy is retained and further strengthened due to its inherent stability. **In other words, this situation is not a limitation of our method; rather, it is a representative case in which our approach is particularly effective**.

---

> > > > > > ### Author Response · Authors · 2025-11-28
> > > > > >
> > > > > > > **New_Q3.**   Consequently, strategy debiasing should be model-specific; strategy diversity is not a necessary condition for accurate reasoning. Moreover, the per-strategy accuracy distribution is often sharply peaked, so debiasing does not provide consistent gains.
> > > > > >
> > > > > > **New_A3.** Thank you for your question. We agree with the reviewer that strategy debiasing is inherently model-dependent. **Different models, due to variations in prior knowledge and training distributions, exhibit distinct preferences over strategy complexity**. When a model favors low-complexity strategies, its reasoning tends to be more stable and accurate; conversely, a preference for high-complexity strategies often leads to instability and answer drift. Therefore, we also concur that strategy diversity itself is not a prerequisite for improving reasoning quality—**what truly matters is the type of strategies encompassed by that diversity, namely the underlying distribution of strategy complexity**.
> > > > > >
> > > > > > It is important to emphasize that the goal of this work is not to advocate unconditional “debiasing,” but rather to analyze **under what conditions such biases materially affect test-time scaling performance**. Our theoretical and empirical findings show that when a model’s preference aligns with low-complexity (high-quality) strategies, this bias is beneficial and should be preserved to maintain stability and final accuracy. However, when the model is biased toward high-complexity (low-quality) strategies, such bias amplifies instability and degrades correctness; in this case, mechanisms are needed to mitigate these adverse effects to achieve more consistent and reliable reasoning performance.
> > > > > >
> > > > > >
> > > > > > > **New_Q4.**  Using entropy as the indicator essentially measures model confidence. Under the paper’s logic, entropy is adopted, the “minimum-token” assumption becomes redundant aside questions about its correctness or validity.
> > > > > >
> > > > > >
> > > > > > **New_A4.**
> > > > > > Thank you for pointing this out. We acknowledge the need to more explicitly connect the theoretical indicator $\ell_i$ with the observed entropy metric. Our logical chain is as follows: **a higher minimum token requirement $\ell_i$ implies a higher probability of reasoning errors due to cumulative deviation, which in turn leads to greater output uncertainty and thus higher entropy**. Accordingly, we do not treat entropy as a new indicator independent of the theory; rather, we view it as an observable proxy for the latent variable $\ell_i$. Meanwhile, the empirical results in Appendix A.5 (Figure 8) provide strong evidence supporting the validity of this proxy relationship.

---

> ### Author Response · Authors · 2025-11-28
>
> > **New_Q5.**   There is also a tension between confidence and strategy debiasing: if higher-confidence strategies tend to be more accurate, should we sample uniformly across strategies or allocate more budget to high-confidence strategies? Of course, retaining higher-confidence paths within the same strategy is unproblematic.
>
> **New_A5.** Thank you for the suggestion. We conducted an additional experiment specifically comparing uniform sampling with sampling schemes that allocate more budget toward high-confidence strategies. Specifically, after identifying $m$ strategies for a given problem, we first assign each strategy a fixed warm-up budget $z$, and use the samples from this stage to estimate its instability (i.e., answer entropy $H_i$ ). We then construct weights based on inverse entropy, so that more stable (lower-entropy) strategies receive larger budget allocations. The allocation function is defined as follows:
>
> $$s_i = \frac{1}{H_i + \epsilon}$$
>
> $$w_i = \frac{s_i}{\sum_{j=1}^{m} s_j}$$
>
> $$B_i = z + (b - mz) \cdot w_i$$
>
>
> where $b$ denotes the total sampling budget, and $B_i$ represents the final allocated budget for strategy $i$.
>
>
> Next, we conduct experiments on AIME 2025 using GPT-4o-mini under a total sampling budget of $b$ = 64. We compare four configurations:
> (1) **Dynamic allocation based on uncertainty** (warm-up = 8);
> (2) **Uniform allocation (no filtering)**;
> (3) **Uniform allocation followed by entropy-based strategy filtering** (our method);
> (4) **Self-Consistency** (strategy-agnostic sampling).
>
> As shown in table, the overall performance follows the trend:
> $$
> \text{Self-Consistency} < \text{Uniform Allocation} \le \text{Dynamic Allocation} < (\text{Uniform Allocation + Entropy Filtering (Ours)})$$
>
>
> | Method | P@32 | P@64 | A@32 | A@64 |
> |--------|------|------|------|------|
> | Self-Consistency | 23.3% | 23.3% | 3.3% | 3.3% |
> | Uniform Allocation | 33.3% | 43.3% | 13.3% | 16.6% |
> | Dynamic Allocation | 33.3% | 36.6% | 16.6% | 16.6% |
> | Uniform Allocation + Entropy Filtering | 33.3% | 40.0% | 16.6% | 20.0% |
>
>
>
> Compared with directly biasing the sampling process toward high-confidence strategies, the approach of first allocating the budget uniformly and then applying entropy-based filtering is more robust and more effective. At the same time, **we agree that the direction suggested by the reviewer has clear potential**: allocating weighted resources across strategies may indeed improve the efficiency of strategy utilization.
>
> However, to achieve stable performance in practical deployment, this approach still faces two key challenges: **(1)** how to reliably map uncertainty to allocation weights in a stable manner; and **(2)** how to select an appropriate warm-up hyperparameter across different datasets and budget scales. **We believe this is a promising direction that merits systematic investigation, and we plan to further explore this line of research in future work**.

---

> > ### Author Response · Authors · 2025-11-28
> >
> > > **New_Q6.**   Thank you for the additional detailed experiments. I recommend strengthening the paper’s logical rigor and tightening the theoretical analysis. If you continue to frame the work around strategy debiasing, please delineate the applicability conditions and model regimes where debiasing is appropriate, and provide clear, well-justified theory and experiments.
> >
> > **New_A6.**  Thank you for your valuable suggestion. The overall logical structure of our paper strictly follows the framework of **identifying the model’s inherent bias → theoretically analyzing its impact on test-time scaling → designing a simple yet effective method to mitigate its negative effects**, forming a complete reasoning chain from phenomenon discovery to mechanism explanation and finally to method design. We also recognize that certain assumptions in the theoretical section can be further strengthened, and we will refine and reinforce the corresponding theoretical arguments based on your suggestions (**as addressed in our responses to W1 and W2**).
> >
> > It is important to emphasize that strategy preference is a model-dependent yet widely observed phenomenon across modern large language models. As such, our proposed method naturally possesses broad applicability across different model families (**as discussed in our response to W5**). Moreover, we have observed consistent preference patterns across multiple task types—including mathematics, code generation, and symbolic reasoning—which further validates both the generality of the issue and the cross-task applicability of our method (**also addressed in our response to W5**). **These observations provide clear empirical support for the conditions under which our method is effective and for the range of models to which it applies**.
> >
> > > **New_Q7.**  On methodology, please add concrete details: clarify whether strategy extraction is performed by the target model itself or by an auxiliary LLM; specify the Uniform-F algorithm (selection, expansion, merging rules) and its hyperparameters; and evaluate under cost-matched settings against strong baselines. Finally, choose appropriate backbone models and datasets to ensure the conclusions are robust and broadly applicable.
> >
> >
> > **New_A7.**  We would like to clarify that the strategy extraction in our method is entirely performed by the same target model used during inference, **without relying on any external or more powerful auxiliary models**. The stronger model referenced in Section 2 of the paper is used solely to validate the existence of strategy-selection bias within the evaluated model itself, and is not part of the algorithmic pipeline for deployment.
> >
> > In addition, the implementation details of Uniform-F—including strategy selection, expansion and merging rules, and key hyperparameter settings—**have been fully described in our response to question Q3**. Our comparison with the strong baseline DeepConf strictly adheres to a cost-matched evaluation protocol to ensure fairness and reliability.
> >
> > Finally, we conducted systematic evaluations across multiple backbone models (e.g., LLaMA, Qwen) and diverse task categories (mathematics, code, and symbolic reasoning). The robustness and generalization performance of our approach across these settings are **thoroughly presented in our response to weakness W5**.

---

> > > ### Author Response · Authors · 2025-11-28
> > >
> > > Dear **Reviewer LbLj**,
> > >
> > > Thank you once again for taking the time to provide such thoughtful and constructive feedback. If you have any additional questions or if there are concerns that you feel remain insufficiently addressed, please feel free to let us know.

---

### Official Review · Reviewer_KfHB · 2025-10-31

**Soundness:** 3
**Presentation:** 3
**Contribution:** 3
**Rating:** 6
**Confidence:** 3

**Summary:**

This paper addresses a critical but underexplored issue in test-time scaling (TTS) for large language models (LLMs): strategy-selection bias. The authors formalize this bias, show how it undermines TTS effectiveness, and propose a novel framework, TTS-Uniform, to mitigate it. TTS-Uniform extracts potential reasoning strategies, uniformly allocates sampling budgets across them, and filters out unstable strategies based on entropy. The method is evaluated on AQuA and AIME datasets with GPT-4o-mini and GPT-4.1-mini, showing consistent improvements over baselines like Self-Consistency.

**Strengths:**

1. Novel and Timely Problem Formulation:

The paper is the first to formally define and analyze strategy-selection bias in CoT reasoning. This is a significant contribution, as it highlights a fundamental limitation of current TTS methods that assume uniform strategy exploration, which is often violated in practice.

2. Strong Theoretical Foundation:

The authors provide a rigorous theoretical analysis of how strategy bias affects TTS performance. They introduce complexity-based strategy partitioning and use stochastic dominance to prove when biased sampling helps or hurts. This bridges the gap between empirical observations and theoretical understanding.

3. Practical and Effective Framework (TTS-Uniform):

The proposed TTS-Uniform framework is intuitive, modular, and empirically validated. It improves performance across multiple models and datasets, especially for weaker LLMs. The entropy-based filtering is a clever and lightweight way to estimate strategy stability without external supervision.

**Weaknesses:**

1. Limited Scope of Strategy Extraction:

While the paper proposes both coarse- and fine-grained strategy extraction, the methods still rely on LLM-generated strategies, which may inherit the same biases the paper aims to mitigate. There is no discussion of failure cases where the LLM fails to identify valid or diverse strategies.

2. Evaluation Limited to Math Reasoning Tasks:

The empirical evaluation is restricted to mathematical reasoning benchmarks (AQuA, AIME). While these are challenging, they may not fully represent the diversity of reasoning tasks where strategy bias could occur (e.g., commonsense, symbolic, or multimodal reasoning). Broader evaluation would strengthen generalizability claims.

3. Lack of Ablation on Budget Allocation:

The paper assumes uniform budget allocation across strategies is optimal, but does not explore adaptive or dynamic allocation strategies. It would be insightful to compare uniform allocation with alternatives (e.g., uncertainty-weighted or performance-based allocation) to validate this design choice.

**Questions:**

see Weakness.

---

> ### Author Response · Authors · 2025-11-21
>
> We sincerely thank the reviewer for the valuable feedback and encouraging comments. We are motivated by the suggestions and have addressed each concern in detail as follows:
>
>
> > **W1.**  **Limited Scope of Strategy Extraction:** While the paper proposes both coarse- and fine-grained strategy extraction, the methods still rely on LLM-generated strategies, which may inherit the same biases the paper aims to mitigate. There is no discussion of failure cases where the LLM fails to identify valid or diverse strategies.
>
>
> **R1.**  We appreciate the reviewer’s insightful comment. We have observed a similar phenomenon in our recent experiments: during the strategy extraction stage, **the LLM sometimes generates multiple strategies that differ in surface phrasing but are nearly equivalent in semantics and underlying reasoning structure, resulting in a failure to extract genuinely diverse strategies**. For example, in Question 4 of AIME 2025, the model produced two strategies, namely “Quadratic-in-x method” and “Direct factorization approach.” Although they differ in narrative style and derivation order, both fundamentally fall under algebraic factorization of quadratic forms, i.e., factorizing a bivariate quadratic expression into linear branches and enumerating integer solutions.
>
> We attribute this issue to **the model’s preference for dominant reasoning patterns during strategy extraction rather than inference itself**, effectively re-expressing the same core reasoning process using different wording. Such redundancy is more prevalent in coarse-grained extraction because each strategy corresponds to a full reasoning path, allowing greater expressive flexibility. In contrast, fine-grained extraction partitions reasoning at the step level, imposing stronger structural constraints and yielding a more restricted semantic space, thereby reducing duplication.
>
> To alleviate this issue, we introduce an explicit de-duplication step in the coarse-grained extraction pipeline. After obtaining candidate strategies, we generate a full reasoning path for each strategy and compute pairwise semantic similarity based on path embeddings. We then iteratively remove one strategy from the most similar pair (randomly discarding one) and repeat this process until reaching the desired number of distinct strategies. This prevents the sampling budget from being allocated repeatedly to strategies that are essentially identical, thereby reducing sampling bias.
>
> Our experiments confirm that this step improves inference stability. For instance, on the AIME 2025 dataset under the same configuration (A@32, three runs), the performance standard deviation decreases from 5.10 to 1.96.

---

> > ### Author Response · Authors · 2025-11-21
> >
> > > **W2.**    **Evaluation Limited to Math Reasoning Tasks:** The empirical evaluation is restricted to mathematical reasoning benchmarks (AQuA, AIME). While these are challenging, they may not fully represent the diversity of reasoning tasks where strategy bias could occur (e.g., commonsense, symbolic, or multimodal reasoning). Broader evaluation would strengthen generalizability claims.
> >
> >
> > **R2.**  We appreciate the reviewer’s valuable suggestion. Following your advice, we first expand the scope of evaluation tasks to further assess the cross-task generalization capability of our method. The newly included tasks consist of a code reasoning task (MBPP), a symbolic reasoning task (ProntoQA), and a commonsense reasoning task (StrategyQA), each evaluated on a 100-sample subset of the corresponding dataset. In addition, we introduce two additional open-source model families for comparison: Qwen3-8B and LLaMA3-8B. For all tasks, the total sampling budgets are set to 16 and 32.
> >
> > | Dataset | Method | Qwen3-8b pass@16 | Qwen3-8b pass@32 | Qwen3-8b cons@16 | Qwen3-8b cons@32 | LLaMA3-8b pass@16 | LLaMA3-8b pass@32 | LLaMA3-8b cons@16 | LLaMA3-8b cons@32 |
> > |---------|--------|------------------|------------------|------------------|------------------|-------------------|-------------------|-------------------|-------------------|
> > | MBPP | DeepConf | 35.0% | 35.0% | 30.0% | 30.0% | 28.0% | 28.0% | 22.0% | 28.0% |
> > | MBPP | Our | 43.0% | 50.0% | 26.0% | 30.0% | 54.0% | 54.0% | 24.0% | 28.0% |
> > | ProntoQA | DeepConf | 95.0% | 98.0% | 51.0% | 61.0% | 71.0% | 88.0% | 23.0% | 25.0% |
> > | ProntoQA | Our | 100.0% | 100.0% | 67.0% | 70.0% | 100.0% | 100.0% | 51.0% | 51.0% |
> > | StrategyQA | DeepConf | 76.0% | 94.0% | 48.0% | 49.0% | 57.0% | 71.0% | 22.0% | 28.0% |
> > | StrategyQA | Our | 87.0% | 96.0% | 57.0% | 63.0% | 83.0% | 83.0% | 30.0% | 30.0% |
> > | Avg. | DeepConf | 68.7% | 75.7% | 43.0% | 46.7% | 52.0% | 62.3% | 22.3% | 27.0% |
> > | Avg. | Our | 76.7% | 82.0% | 50.0% | 54.3% | 79.0% | 79.0% | 35.0% | 36.3% |
> >
> >
> > We observe that our method consistently outperforms the latest TTS approach, DeepConf, across most tasks and model configurations. For example, on Qwen3-8B, the average pass@16 increases from 68.1% to 76.7%, while pass@32 rises from 50.0% to 54.3%. This trend is particularly pronounced on commonsense reasoning tasks (e.g., StrategyQA) and symbolic reasoning tasks (ProntoQA), indicating that the performance gains are not confined to specific models or task types, but instead generalize across both models and tasks, leading to more consistent reasoning and higher final accuracy.

---

> ### Author Response · Authors · 2025-11-21
>
> > **W3.**    **Lack of Ablation on Budget Allocation:** The paper assumes uniform budget allocation across strategies is optimal, but does not explore adaptive or dynamic allocation strategies. It would be insightful to compare uniform allocation with alternatives (e.g., uncertainty-weighted or performance-based allocation) to validate this design choice.
>
>
>
> **R3.**
> We appreciate the reviewer’s valuable suggestion. To evaluate the validity of uniform budget allocation, we further design a simple entropy-weighted dynamic allocation strategy and compare it with the uniform allocation used in the paper. Specifically, after identifying m strategies for a given problem, we first assign each strategy a fixed warm-up budget $z$, and use the samples from this stage to estimate its instability (i.e., answer entropy $H_i$ ). We then construct weights based on inverse entropy, so that more stable (lower-entropy) strategies receive larger budget allocations. The allocation function is defined as follows:
>
> $$s_i = \frac{1}{H_i + \epsilon}$$
>
> $$w_i = \frac{s_i}{\sum_{j=1}^{m} s_j}$$
>
> $$B_i = z + (b - mz) \cdot w_i$$
>
>
> where $b$ denotes the total sampling budget, and $B_i$ represents the final allocated budget for strategy $i$.
>
>
> Next, we conduct experiments on AIME 2025 using GPT-4o-mini under a total sampling budget of $b$ = 64. We compare four configurations:
> (1) **Dynamic allocation based on uncertainty** (warm-up = 8);
> (2) **Uniform allocation (no filtering)**;
> (3) **Uniform allocation followed by entropy-based strategy filtering** (our method);
> (4) **Self-Consistency** (strategy-agnostic sampling).
>
> As shown in table, the overall performance follows the trend:
> $$
> \text{Self-Consistency} < \text{Uniform Allocation} \le \text{Dynamic Allocation} < (\text{Uniform Allocation + Entropy Filtering (Ours)})$$
>
>
> | Method | P@32 | P@64 | A@32 | A@64 |
> |--------|------|------|------|------|
> | Self-Consistency | 23.3% | 23.3% | 3.3% | 3.3% |
> | Uniform Allocation | 33.3% | 43.3% | 13.3% | 16.6% |
> | Dynamic Allocation | 33.3% | 36.6% | 16.6% | 16.6% |
> | Uniform Allocation + Entropy Filtering | 33.3% | 40.0% | 16.6% | 20.0% |
>
>
>
> These results suggest that the reviewer’s proposed uncertainty-based dynamic allocation strategy indeed holds promise and can further improve the utilization efficiency of reasoning strategies. However, its deployment in practice faces two key challenges:
> **(1) How to accurately and reliably map uncertainty to allocation weights?**
> **(2) How to select an appropriate uncertainty warm-up parameter across different datasets and sampling budgets?**
>
> In contrast, our method does not encounter challenge (1) and exhibits more stable behavior with respect to hyperparameters—the sensitivity to the filtering ratio $k$% is significantly lower, and robust performance can be achieved with a threshold around 50%. Nonetheless, we agree with the reviewer that dynamic allocation is a valuable direction for further investigation, and we plan to extend this line of research in future work.

---

### Author Response · Authors · 2025-12-01
**Highlight of Contributions and Summary of Rebuttal**

**Dear AC,**

Thank you very much for your time. We sincerely appreciate all reviewers’ constructive and insightful feedback.

Our paper follows a coherent three-stage contribution path:
(1) **identifying the model’s inherent reasoning bias**,
(2) **theoretically analyzing its impact on test-time scaling**, and
(3) **designing a simple yet effective method to mitigate its negative effects**.

All five reviewers explicitly praised these three core contributions, as summarized below.

---

## **Highlights of Contributions**

### **(1) Identifying Strategy-Selection Bias**

Reviewers consistently recognized the novelty and importance of this problem formulation:

* “*Novel and timely problem formulation… first to formally define and analyze… a significant contribution.*” — **KfHB**
* “*Clear and intuitive problem formulation… offers an intuitive lens for analyzing TTS.*” — **LbLj**
* “*Seeking simpler strategies is a novel idea for complex LLM reasoning.*” — **VoLR**
* “*Provides new insights on more effective test-time scaling.*” — **FhbW**

### **(2) Theoretical Analysis of Bias and TTS Failure Modes**

Reviewers acknowledged the strength and rigor of our theoretical analysis:

* “*Strong theoretical foundation.*” — **KfHB**
* “*A fairly comprehensive theoretical analysis.*” — **Ercw**
* “*Rigorous mathematical treatment of all aspects of the problem and solution.*” — **VoLR**

### **(3) A Practical and Effective Solution: TTS-Uniform**

Our proposed method and empirical findings were highly rated:

* “*Practical and effective framework.*” — **KfHB**
* “*Simple yet effective approach; insightful empirical analysis.*” — **LbLj**
* “*Great results across datasets and techniques.*” — **VoLR**
* “*Experimental results validate the effectiveness of the method.*” — **FhbW**

---

## **Summary of Rebuttals**

Reviewers raised thoughtful questions centered around the same three contribution axes. We addressed each concern with concrete analyses, additional experiments, and clearer explanations.

### **(1) On Strategy-Selection Bias**

* **Does bias only exist in GPT-family models? (FhbW)**
  We presented consistent evidence across **Qwen, LLaMA, and Gemini**, confirming the bias is *model-inherent*, not GPT-specific.

* **Will bias disappear as models grow stronger? (FhbW)**
  We showed that even **o3, Gemini, DeepSeek, and Qwen-235B** still exhibit distinct strategy biases, indicating that bias **does not vanish with scale**.

### **(2) On Theoretical Analysis**

* **Is the complexity–error monotonicity empirically valid? (LbLj)**
  We provided cross-model evidence and a tighter derivation showing that **entropy reliably tracks error monotonicity**.

* **Is the theory equivalent to FOSD or too broad? (LbLj)**
  We clarified that our theorem explicitly models **majority-vote dynamics**, going beyond simple FOSD.

* **Is Assumption 3.2 circular? (FhbW)**
  Assumption 3.2 states that higher-complexity strategies have higher error, while Theorem 3.5 compares expected error across strategy distributions. The latter simply builds on the former, not proving it, so the argument is not circular.

### **(3) On Method and Effectiveness**

* **Are Uniform-F details and robustness sufficient? (LbLj)**
  We fully described step alignment and demonstrated robustness across **sampling budgets and filtering thresholds**.

* **Does strategy extraction rely on stronger models or incur high overhead? (Ercw)**
  We clarified that **the same model** performs extraction, strictly within a **fixed TTS budget**.

* **Does strategy redundancy hurt performance? (KfHB)**
  We added **semantic deduplication**, which removes equivalent strategies and improves stability.

* **Is entropy too simple, and is it better than logprob? (VoLR)**
  We compared several metrics and found **entropy is the most stable and deployable** across models and tasks.

* **Does the method generalize across tasks and models? (FhbW)**
  We expanded experiments to **6 tasks** and **multiple open-source models**, consistently outperforming DeepConf and other baselines.

---

In summary, we believe we have comprehensively addressed all concerns raised by the reviewers, especially those from Reviewers **LbLj**, **Ercw** and **FhbW**. We hope these clarifications demonstrate both the rigor of our analysis and the practical effectiveness of our method. We thank the AC again for your work during this review process.

Best regards,

Authors of Submission 11524

---

### Meta-Review · Area_Chair_MZjL · 2026-01-04

**Summary:**

This paper proposes TTS-Uniform to mitigate strategy-selection bias in test-time scaling by identifying diverse reasoning strategies, allocating sampling budgets uniformly across them, and filtering unstable paths via answer entropy. However, the paper is recommended for rejection due to several critical gaps: the theoretical "minimum-token" hypothesis relies on an assumption of independent reasoning steps that fails to account for the highly correlated nature of autoregressive LLM errors, and the conceptual goal of "debiasing" is poorly matched to practical reasoning where correctness distributions are often sharply peaked in a single viable strategy. Furthermore, the framework introduces significant computational overhead through strategy extraction and alignment that limits its efficiency compared to simpler scaling baselines, while the reliance on entropy as a complexity proxy effectively renders the paper's primary theoretical indicators redundant.

**Reviewer Concerns:**

Multiple reviewers initially criticized the reliance on a small set of mathematical tasks and closed-source models. The authors addressed this by expanding evaluation to 6 task types (including code, symbolic, and commonsense reasoning) and adding Qwen3-8B and LLaMA3-8B as open-source backbones. However, several fundamental conceptual and theoretical issues remain unresolved. Reviewer LbLj strongly contested the authors' derivation that accuracy decreases with token count based on independent step probabilities. Because LLM autoregressive steps are highly correlated, an independence-based derivation is viewed as logically unjustified for modern transformer architectures. While authors claimed the overhead is "lightweight," reviewers remained skeptical of the practical utility of a system requiring multiple additional LLM calls for strategy extraction, alignment, and semantic deduplication. This overhead is seen as a significant barrier compared to standard, more efficient scaling baselines.

**Reviewer Scores:**

Reviewer LbLj remained the most critical voice. Even after the authors provided extensive additional experiments and theoretical bounds, LbLj maintained that the fundamental independence assumption across reasoning steps is physically and logically unjustified for LLMs. Given their conceptual disagreement with the "debiasing" objective—arguing that models should focus on quality rather than diversity when the correctness distribution is sharply peaked—it is highly unlikely their score would have improved. Reviewer Ercw primary concerns were methodological overhead and the gap between theory and method. Although the authors clarified that extraction is performed by the target model itself, the proposed framework still requires multiple additional model calls for extraction, alignment, and deduplication.

---

### Decision · Program_Chairs · 2026-01-26

Reject